# An interbacterial cysteine protease toxin inhibits cell growth by targeting type II DNA topoisomerases GyrB and ParE

**Pin-Yi Song[1,2☾], Chia-En Tsai[1,2☾], Yung-Chih Chen[2], Yu-Wen Huang[2], Po-Pang Chen[2,3], Tzu-Haw Wang[4], Chao-Yuan Hu[2], Po-Yin Chen[1,2], Chuan Ku[4,5], Kuo-Chiang Hsia[1,2,3], See-Yeun Ting[1,2,5]***

1 Molecular and Cell Biology, Taiwan International Graduate Program, Academia Sinica and National Defense Medical Center, Taipei, Taiwan, 2 Institute of Molecular Biology, Academia Sinica, Taipei, Taiwan, 3 Institute of Biochemistry and Molecular Biology, College of Life Sciences, National Yang Ming Chiao Tung University, Taipei, Taiwan, 4 Institute of Plant and Microbial Biology, Academia Sinica, Taipei, Taiwan, 5 Genome and Systems Biology Degree Program, National Taiwan University, Taipei, Taiwan

☾ These authors contributed equally to this work.
* syting@gate.sinica.edu.tw

## Abstract

Bacteria deploy a diverse arsenal of toxic effectors to antagonize competitors, profoundly influencing the composition of microbial communities. Previous studies have identified an interbacterial toxin predicted to exhibit proteolytic activity that is broadly distributed among gram-negative bacteria. However, the precise mechanism of intoxication remains unresolved. Here, we demonstrate that one such protease toxin from *Escherichia coli*, Cpe1, disrupts DNA replication and chromosome segregation by cleaving conserved sequences within the ATPase domain of type II DNA topoisomerases GyrB and ParE. This cleavage effectively inhibits topoisomerase-mediated relaxation of supercoiled DNA, resulting in impaired bacterial growth. Cpe1 belongs to the papain-like cysteine protease family and is associated with toxin delivery pathways, including the type VI secretion system and contact-dependent growth inhibition. The structure of Cpe1 in complex with its immunity protein reveals a neutralization mechanism involving competitive substrate binding rather than active site occlusion, distinguishing it from previously characterized effector-immunity pairs. Our findings unveil a unique mode of interbacterial intoxication and provide insights into how bacteria protect themselves from self-poisoning by protease toxins.

## Introduction

Bacteria in natural environments exist within complex communities, where interactions—both cooperative and antagonistic—are pivotal to their survival. A key aspect of bacterial competition is the acquisition of limited resources, which has driven the evolution of a wide array of interbacterial antagonistic mechanisms, including the

**Data availability statement:** All relevant data are within the paper and its Supporting Information files. The coordinates and structure factors for the Cpe1$_{tox}$–Cpi1 complex have been deposited in the Protein Data Bank, with accession code 9L5U. Raw proteomic data have been deposited in the Zenodo database (https://doi.org/10.5281/zenodo.15361709).

**Funding:** This research was supported by Academia Sinica (AS-CDA-112-L05 to S.-Y.T., AS-IVA-112-L05 to K.-C.H., AS-CDA-110-L01 to C.K.) and the National Science and Technology Council, Taiwan (111-2311-B-001-006-MY2, 112-2311-B-001-020, and 113-2628-B-001-008-MY3 to S.-Y.T., 112-2320-B-001-006-MY3 to K.-C.H., 111-2611-M001-008-MY3 to C.K.). The funders had no role in study design, data collection and analysis, decision to publish, or preparation of the manuscript.

**Competing interests:** The authors have declared that no competing interests exist.

**Abbreviations:** ABC, ATP-binding cassette; AF3, AlphaFold3; BLI, biolayer interferometry; CDI, contact-dependent growth inhibition; CFUs, colony-forming units; Cpe1, cysteine protease effector 1; FDR, false discovery rate; PAE, predicted assigned error; PCR, polymerase chain reaction; PLCPs, papain-like cysteine proteases; PSMs, peptide spectrum matches; T4SS, type IV secretion system; T6SS, type VI secretion system.

production of antibiotics, antimicrobial peptides, and diffusible proteinaceous toxins [1,2]. More recently, sophisticated mechanisms for distinguishing kin and antagonizing competitors have been identified [3–8]. Among these, gram-negative bacteria utilize specialized secretion systems, such as contact-dependent growth inhibition (CDI) [3], the type IV secretion system (T4SS) [7], and the type VI secretion system (T6SS) [5], to translocate toxins into neighboring cells in a contact-dependent manner. In contrast, Gram-positive bacteria rely on the type VII secretion system (also known as the Esx secretion system) for similar antagonistic interactions [9]. These secretion systems allow pathogenic bacteria to establish dominance in polymicrobial infections and enable commensal bacteria to establish compatibility within microbial consortia [10–12].

Interbacterial toxins delivered via these systems enhance the competitive fitness of the producing bacteria by inhibiting or killing competitors. Depending on their biochemical properties, the impact of these effectors on target cells can vary considerably. For instance, a wide variety of antibacterial toxins induce cell lysis by disrupting the cellular membrane or degrading cell wall peptidoglycans [9,13–17]. Pore-forming toxins exert their toxicity by inserting themselves into the cellular membrane and causing its depolarization [18–20]. Some toxins are translocated into the cytoplasm, where they compromise DNA/RNA integrity or deplete the cellular energy resources of target cells [21–26]. Post-translationally modified toxins have been found to target essential proteins involved in bacterial cell division and protein synthesis [27–30]. This broad spectrum of targets allows the toxins to act across phylogenetically diverse bacteria. Most of the intoxication mechanisms of these antibacterial toxins are inherently indiscriminate. Therefore, bacteria encode specific immunity proteins adjacent to their cognate toxin genes to prevent self-intoxication or harm to kin cells. These immunity proteins typically neutralize toxins by occluding their active sites [31]. In some cases, other neutralization mechanisms, such as enzymatic antagonism of effector activity or structural disruption of the effector domain, have also been reported [27,32].

Although significant progress has been made in identifying novel interbacterial toxins in recent years, a detailed characterization of their intoxication mechanisms remains lacking [33–35]. For example, comparative genomic, advanced genetic screening, and structure-based searching approaches have uncovered novel effectors with conserved catalytic residues indicative of protease and peptide amidase activities, which inhibit bacterial growth during interbacterial competition [36–38]. However, their natural substrates and precise biochemical activities have yet to be elucidated. In this study, we describe an interbacterial toxin that targets the essential type II DNA topoisomerases GyrB and ParE via papain-like cysteine protease (PLCP) activity. This toxin is phylogenetically widespread and shares hallmarks of interbacterial effectors secreted via the T6SS and CDI pathways. Homologs of this toxin are similar to PLCPs associated with ATP-binding cassette (ABC) transporters involved in bacterial protein export. However, their role in interbacterial antagonism, specifically in targeting and damaging essential proteins, has not been elucidated. Interestingly, our structural analyses further

reveal that protection against the PLCP toxins is conferred by cognate immunity proteins, which block substrate binding rather than occlude the active site, thereby inactivating the toxicity. Our findings not only expand our understanding of protease effectors in interbacterial conflict, but also uncover a previously undescribed mode of immunity protection.

## Results

### Papain-like cysteine protease antibacterial effectors are found in different gram-negative bacteria

Previous studies on T6SS–associated protease effectors, such as EvpQ from *Edwardsiella piscicida* and TsaP from *Escherichia coli*, have predicted their classification within the family of PLCPs [36,37]. The defining feature of this protease family is a conserved catalytic triad, consisting of a cysteine (Cys) thiol group, a histidine (His) imidazolium ring, and a third residue—either aspartate (Asp) or asparagine (Asn)—that orients and activates the imidazolium ring for peptide bond hydrolysis (Fig 1a). PLCPs in bacteria have been well characterized for their roles as virulence factors in both animal and plant infections [39–41], as well as for functioning as maturation proteases in a variety of ABC transporters, where they specifically recognize and cleave signal peptides before substrate export [42–44]. However, their function as antibacterial toxins has yet to be fully understood [33].

To explore the prevalence of PLCPs involved in interbacterial antagonism, we conducted a homology search against the National Center for Biotechnology Information (NCBI) non-redundant database using PSI–BLAST [45], employing the cysteine protease domains of EvpQ and TsaP as queries [36,37]. Homologous genes with more than 10% sequence identity were retrieved, and their gene neighborhoods were analyzed to infer potential roles in interbacterial competition. The low sequence identity threshold was selected to ensure identification of both EvpQ- and TsaP-like proteins. Consequently, we identified 155 unique genes encoding PLCP domains associated with either the T6SS or CDI pathways. Of these, 70 were found to be associated with the T6SS, where they appeared in diverse configurations—either fused to or encoded as separate genes downstream of various toxin carriers, including PAAR, VgrG, and Rhs [46,47]—and the remaining 85 were linked to the CdiA carrier protein in the CDI pathway [48] (Figs 1b, S1a, and S1 Table). Multiple sequence alignments of these homologs, along with representative PLCPs from ABC transporters (*e.g.,* ComA and LahT), confirmed the presence of the conserved catalytic triad (Cys, His, Asp/Asn) [49,50] (Fig 1c).

Phylogenetic analysis of the identified protease domain sequences revealed close similarities to EvpQ and TsaP [36,37] (Figs 1d and S2). These toxin homologs are distributed across Phylum Pseudomonadota (previously designated as Proteobacteria), including Alpha-, Beta-, and Gamma-proteobacteria, and they are found in bacteria across a range of ecological niches, including within animal and insect hosts and in the environment. Additionally, 15 T6SS–associated homologs were identified in distantly related phyla, such as Acidobacteriota, Actinomycetota, Chloroflexota, Cyanobacteriota, and Myxococcota (Figs 1d, S1–S2, and S1 Table). Although we detected no clear correlation between phylogeny, taxonomy, and toxin-delivery pathway, the phylogenetic similarity of these homologs from diverse phyla implies that horizontal gene transfer may have contributed to the evolutionary dissemination of PLCP toxins.

To prevent self-intoxication, toxin-producing bacteria carry immunity proteins, typically encoded adjacent to the toxin genes in the genome [31]. Consistent with this scenario, we found that the majority of the identified PLCP toxin genes are followed in the respective genomes by putative immunity genes (Figs 1b and S1b). The putative immunity proteins range from 79 to 239 amino acids and exhibit considerable sequence divergence, likely reflecting their specificity towards their cognate PLCP toxins. Supporting this idea, the sequence identity between each immunity gene and its corresponding PLCP toxin is positively correlated, evidencing their coevolutionary nature (S1 Table). Together, these findings suggest that PLCP toxins, along with their cognate immunity proteins, are found in various gram-negative bacterial species. For clarity in this study, we have designated the identified PLCP toxin as Cpe1 (cysteine protease effector 1) and the cognate immunity protein as Cpi1 (Fig 1b, 1d).

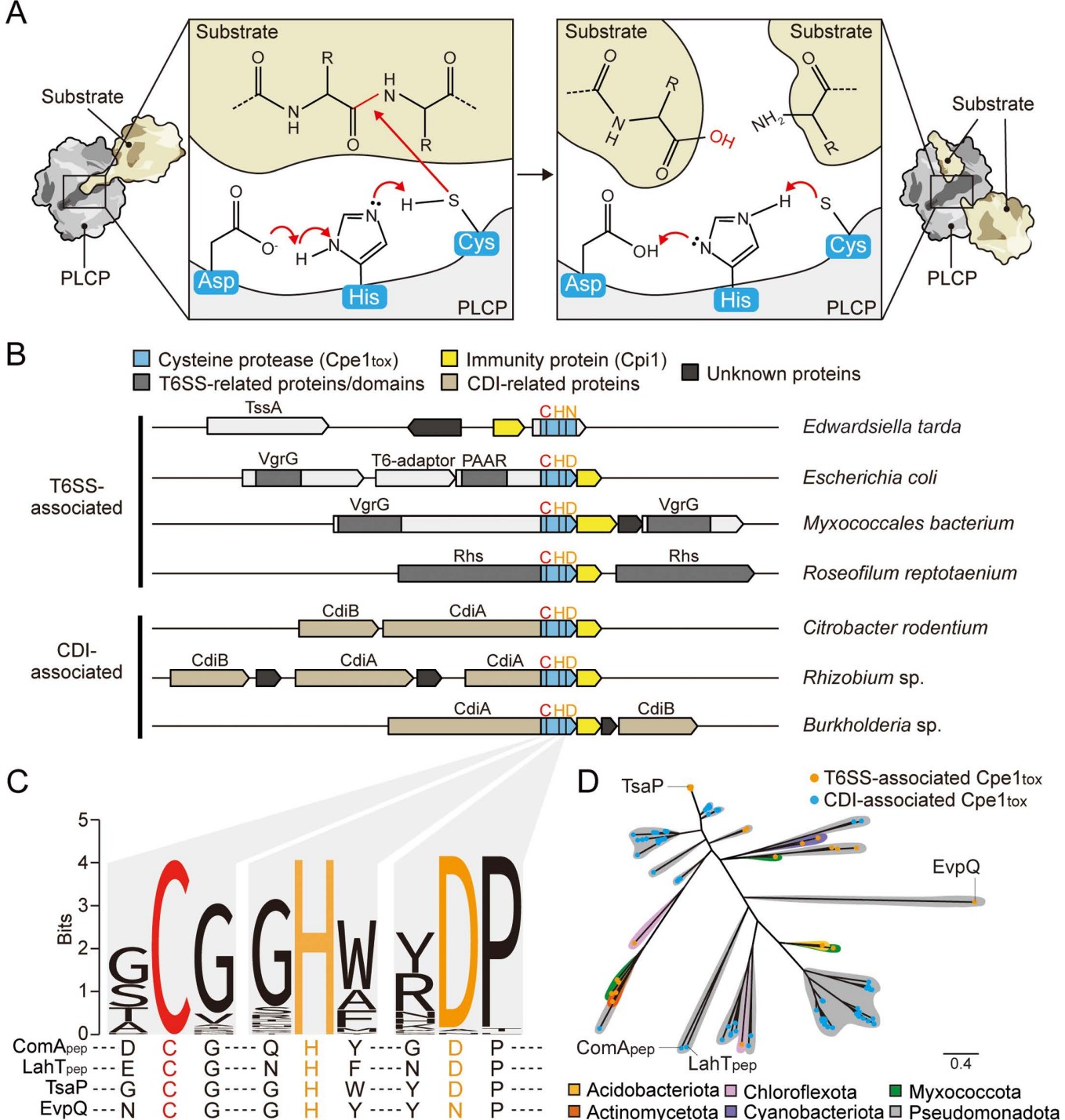

**Fig 1. PLCP toxins are found in distinct gram-negative species and are associated with interbacterial toxin delivery pathways. (a)** The enzymatic reaction mechanism of papain-like cysteine proteases (PLCPs). The catalytic triad of residues in PLCPs, composed of cysteine (Cys), histidine (His), and aspartate (Asp), facilitates peptide bond cleavage. Histidine and aspartate form a proton-withdrawing group that activates the cysteine residue, converting it into a nucleophile. The activated cysteine attacks the carbonyl carbon of the peptide bond, leading to hydrolysis and bond cleavage. **(b)** Scaled depiction of representative genomic loci from the indicated species (*Edwardsiella tarda*, *Escherichia coli*, *Myxococcales bacterium*, *Roseofilum reptotaenium*, *Citrobacter rodentium*, *Rhizobium* sp., *Burkholderia* sp.) encoding predicted PLCP domain-containing proteins (sky blue) and putative

PLOS Biology

cognate immunity proteins (yellow). Regions encoding N-terminal domains of T6SS-related proteins (PAAR, VgrG, Rhs) are shaded in gray. CDI-related proteins (CdiA, CdiB) are depicted in light brown. Black bars designate locations of bases encoding the catalytic triads of PLCPs. Cysteine residues targeted for mutagenesis in this study are depicted in red. **(c)** Sequence logo generated from alignments of PLCP domains associated with bacterial contact-dependent antagonism pathways (the locus tag numbers for genes encoding the sequences we used are provided in S1 Table). Sequences from previously characterized PLCPs (ComA$_{pep}$ and LahT$_{pep}$) and recently identified T6SS-associated PLCP toxins (EvpQ and TsaP) are shown below for reference. **(d)** Maximum likelihood phylogeny of PLCP effectors visualized as a radiating phylogenetic tree. Taxonomy is indicated by colored shadows, and delivery pathways are indicated by colored tips. The scale bar represents the average number of substitutions per site. The original tree file used to generate this figure is available as S2 Data. See also S2 Fig.

### The structure of the Cpe1–Cpi1 complex from *Escherichia coli* reveals a distinct toxin-neutralization mechanism

Pioneering studies by Li and colleagues and Lu and colleagues have demonstrated that bacteria, such as *E. piscicida* and *E. coli*, utilize T6SS–associated Cpe1 effectors to mediate interbacterial antagonism [36,37]. However, the precise mechanisms underlying Cpe1-induced toxicity remain unclear. To elucidate the mode of action of Cpe1, we leveraged the phylogenetic analysis described above to identify a T6SS-associated Cpe1 from *Escherichia coli* strain ATCC 11,775 (Fig 1b, S1 Table) and employed structural biology analysis to investigate its toxic activity. Notably, overexpression of the Cpe1 toxin domain (Cpe1$_{tox}$) in lab model *E. coli* strain BL21 (DE3) for protein production resulted in growth inhibition, yielding insufficient amounts of purified protein for structural analyses (Fig 2a). Time-lapse microscopy revealed that this growth inhibition was accompanied by reduced cell proliferation and continued elongation, consistent with the proposed cyto-toxic activity of Cpe1 [37] (Fig 2b and S1–S2 Videos). To circumvent this limitation, we co-purified Cpe1$_{tox}$ with its cognate immunity protein, Cpi1, which allowed us to obtain well-diffracting crystals (S3a Fig). Using selenomethionine-substituted protein, we determined the crystal structure of the Cpe1$_{tox}$–Cpi1 complex at a resolution of 1.87 Å (Fig 2c and S2 Table).

In our crystal structure, Cpe1 features a six-stranded antiparallel β-sheet flanked by five α-helices on either side, forming the typical α/β core that is characteristic of the PLCP family [51]. The catalytic triad—Cys, His, Asp—is clearly evident and aligns with prior functional predictions [33,51] (Fig 2c). Despite sharing limited sequence identity with ABC transporter-associated proteases, a structural homology search using the DALI server identified ComA (Z-score = 16.3; RMSD of 1.9 Å across 121 aligned Cα atoms) and LahT (Z-score = 15.9; RMSD of 1.8 Å across 117 aligned Cα atoms) as the closest homologs [50,52,53] (Fig 2d). A structural comparison revealed a strong resemblance between Cpe1, ComA, and LahT. Superimposing the Cα atoms of the catalytic residues in Cpe1 and LahT indicated minimal structural deviation, with RMSD values of 0.583 Å (Cys), 0.843 Å (His), and 0.711 Å (Asp). The catalytic triad of ComA exhibited a slight shift relative to Cpe1, with RMSD values of 2.444 Å (Cys), 1.812 Å (His), and 1.680 Å (Asp) (Fig 2d). Nonetheless, the overall spatial arrangement of Cpe1's catalytic triad remains highly conserved. These structural insights reinforce the predicted protease function of Cpe1, indicating that its proteolytic activity relies on the common catalytic thiol mechanism (Fig 1a).

In line with our structural data, mutational analyses revealed that the substitution of the catalytic cysteine with alanine (Cpe1$_{tox}$$^{C362A}$) abolished Cpe1-mediated toxicity (Fig 2a). Moreover, the catalytically inactive Cpe1 mutant failed to induce the cell elongation phenotype observed for wild-type Cpe1, further demonstrating that Cpe1 toxicity is linked to its PLCP activity (Fig 2b and S3 Video). To assess if the catalytic activity of Cpe1 is important for interbacterial antagonism, we performed a growth competition assay using an *E. coli* strain (ATCC-11775) harboring the wild-type Cpe1 toxin and an active T6SS as the donor strain, competing against a recipient *E. coli* model strain (MG1655). Notably, recipient cells exhibited a significant fitness defect when co-cultured with donor cells expressing wild-type Cpe1 (Fig 2e). However, this competitive advantage was markedly reduced when donor cells expressed the inactive Cpe1 (Cpe1$_{tox}$$^{C362A}$). These findings demonstrate that the catalytic activity of Cpe1 plays a crucial role in mediating interbacterial competition.

Interestingly, although Cpi1 binds to Cpe1 and they share a large interface (1,050 Å$^2$), our crystal structure reveals that Cpi1 does not sterically occlude the active site of Cpe1. Instead, Cpi1 primarily interacts with a shallow hydrophobic concave region on the surface of Cpe1 that is located adjacent to the catalytic pocket (Fig 2c, 2f). This binding region consists

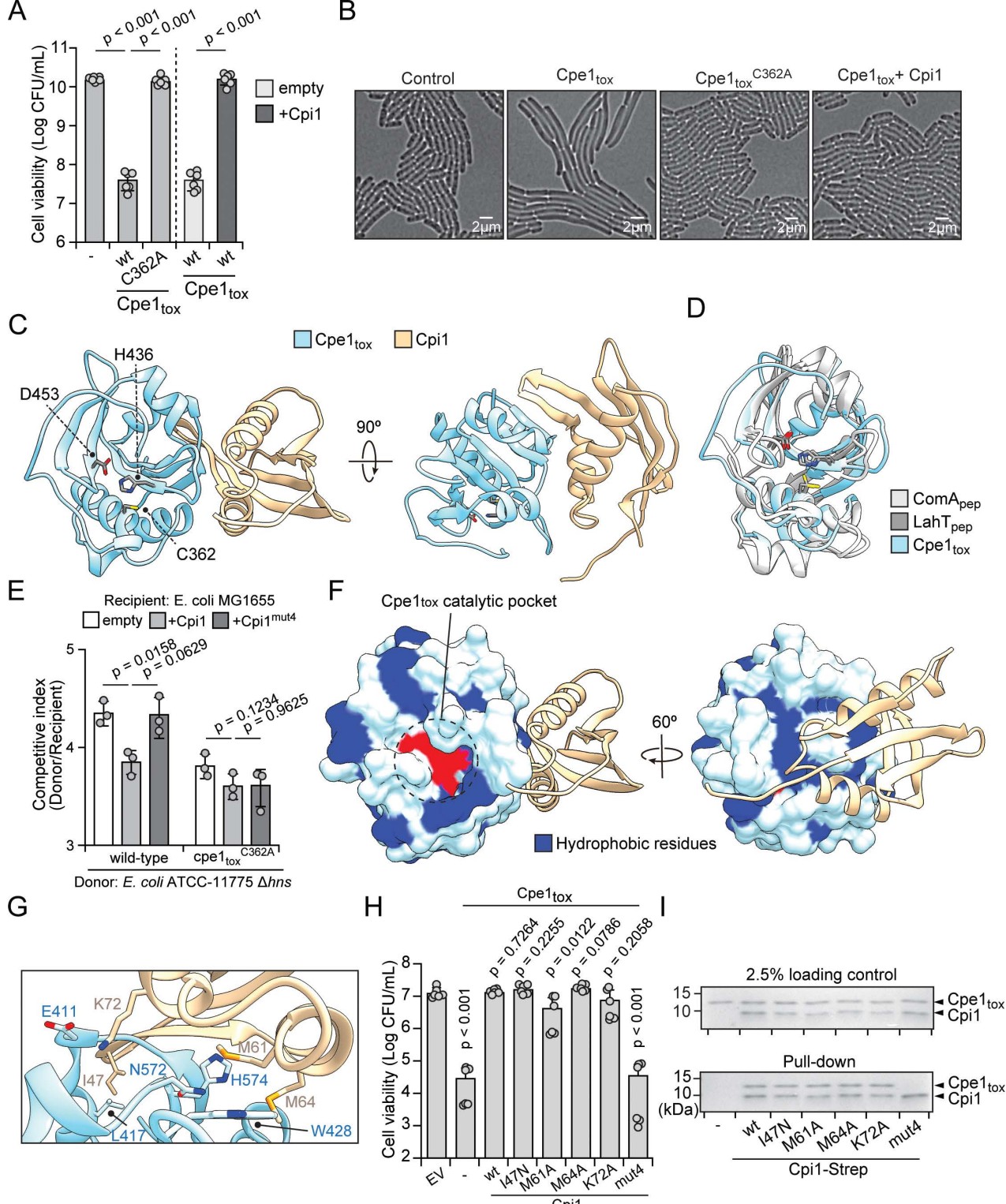

**Fig 2. Cpe1_tox-Cpi1 is a functional effector-immunity pair. (a)** Viable *E. coli* cells recovered from plating cultures carrying plasmids expressing the indicated proteins on inducing media. *E. coli* carrying plasmid-borne Cpi1 or an empty vector was included to validate suppression of Cpe1 toxicity. Data are shown as the mean ± SD; *n* = 6. **(b)** Representative micrographs of *E. coli* cells expressing empty vector, Cpe1_tox, Cpe1_tox^C362A, or co-expressing

Cpe1$_{tox}$ and Cpi1. Frames were acquired 2 h after induction of protein expression. Scale bar, 2 μm. **(c)** Ribbon diagram representation of the X-ray crystal structure of Cpe1$_{tox}$ (sky blue) in complex with Cpi1 (light gold). Residues of the catalytic triad of Cpe1$_{tox}$ are indicated. **(d)** Structural alignments of Cpe1$_{tox}$ with previously characterized PLCPs: ComA$_{pep}$ (PDB: 3K8U, light gray) and LahT$_{pep}$ (PDB: 6MPZ, dark gray). **(e)** Bacterial competition experiments were conducted on a solid agar over a 4-h period between the indicated donor and recipient strains. *E. coli* ATCC-11775 *Δhns* served as the donor strain, whereas *E. coli* MG1,655 was used as the recipient strain. The *Δhns* mutation in the donor strain was introduced to up-regulate T6SS activity, as described in previous studies [102–104]. Data are shown as the mean ± SD; $n = 3$. **(f)** Depiction of Cpe1$_{tox}$ illustrating the Cpe1$_{tox}$–Cpi1 interaction at the exosite formed by hydrophobic residues. **(g)** Magnified view of the interface between Cpe1$_{tox}$ and Cpi1, highlighting the interacting residues on Cpi1 positioned near the hydrophobic concave surface of Cpe1$_{tox}$. **(h)** Viable *E. coli* cells recovered from *E. coli* co-expressing wild-type Cpe1$_{tox}$ and the indicated Cpi1 variants. Data are shown as the mean ± SD; $n = 6$. **(i)** *In vitro* pull-down assay of Cpe1$_{tox}$ and Cpi1. Cpe1$_{tox}$ proteins were pulled down by strep-tagged Cpi1 variants and immobilized on streptavidin resin. Bound proteins were separated by SDS–PAGE followed by Coomassie brilliant blue staining. The data shown in **(a)**, **(e)**, and **(h)** are from a representative experiment of at least three independent experiments. *P* values were calculated with Student *t* test to assess differences in viability among populations **(a, h)**, and to evaluate statistically significant differences in the competitive indices of each donor strain against the specified recipients **(e)**. The data underlying this figure are available in S1 Data and S1 Raw Images. See also S3 Fig, S2 Table, and S1–S4 Videos.

of a cluster of non-polar residues—L387, M390, A408, M426, W428, and V434—the hydrophobic properties of which are conserved across many T6SS- and CDI-associated Cpe1 toxins (S1a Fig). We did not identify any enzymatic domain in or significant structural homologs of Cpi1, suggesting that its neutralization mechanism depends on binding to the hydrophobic concave region of Cpe1 rather than occlusion of its active site.

To examine the importance of the Cpe1–Cpi1 interaction, we assessed the ability of Cpi1 variants displaying impaired Cpe1 binding to protect against toxicity. By scanning the Cpe1–Cpi1 interface, we identified four candidate residues (I47, M61, M64, and K72) in Cpi1, where substitutions were predicted to disrupt the Cpe1–Cpi1 interaction (Fig 2g). Single-residue substitutions did not significantly affect the ability of Cpi1 to inhibit Cpe1 toxicity, as these variants retained their immunity function and still bound Cpe1 in co-immunoprecipitation assays (Fig 2h, 2i). However, when we mutated all four residues (Cpi1$^{mut4}$), the resulting mutant lacked the ability to neutralize Cpe1 toxicity and failed to bind Cpe1 *in vitro*, despite being expressed at levels comparable to that of wild-type protein (Figs 2h, 2i, and S3b). In addition, the Cpi1 quadruple mutant did not provide full protection during interbacterial competition (Fig 2e). These findings indicate that the interaction between Cpe1 and Cpi1 is critical for the immunity function of Cpi1 and that disruption of this interaction abrogates its protective effect (Figs 2a, 2b, and S4 Video).

## The hydrophobic concave surface is critical for Cpe1 toxicity

Our structural analysis revealed that Cpe1 shares significant similarities with the cysteine protease domains of ComA and LahT [50,52] (Fig 2d). Previous biochemical and structural studies of these proteases have shown that, similar to Cpe1, they also possess hydrophobic concave surfaces adjacent to their catalytic pockets, which are critical for specific recognition of protein substrates [50,52] (S4 Fig). Previous studies have also shown that blocking these hydrophobic regions with high-affinity inhibitors leads to protease dysfunction and impaired substrate transport [50,54]. Given that Cpi1 neutralizes Cpe1 toxicity by interacting with its hydrophobic concave region (Fig 2f), we hypothesized that Cpe1 might employ a substrate recognition mechanism analogous to that of ComA and LahT. To test this hypothesis, we introduced mutations into hydrophobic residues within the concave region of Cpe1 (*i.e.,* L387, M426, W428) and assessed their impact on toxicity (Fig 2f). Supporting our hypothesis, ectopic expression of the Cpe1 variants in model *E. coli* resulted in a marked reduction in toxicity compared to wild-type Cpe1, without affecting protein expression levels (Fig 3a). These findings support the notion that the hydrophobic concave region is important for the cytotoxic activity of Cpe1 and may also contribute to substrate recognition.

To further investigate if the hydrophobic concave region of Cpe1 is involved in substrate binding, we carried out site-specific photo-crosslinking experiments, focusing on the residues for which mutations abolished Cpe1-mediated

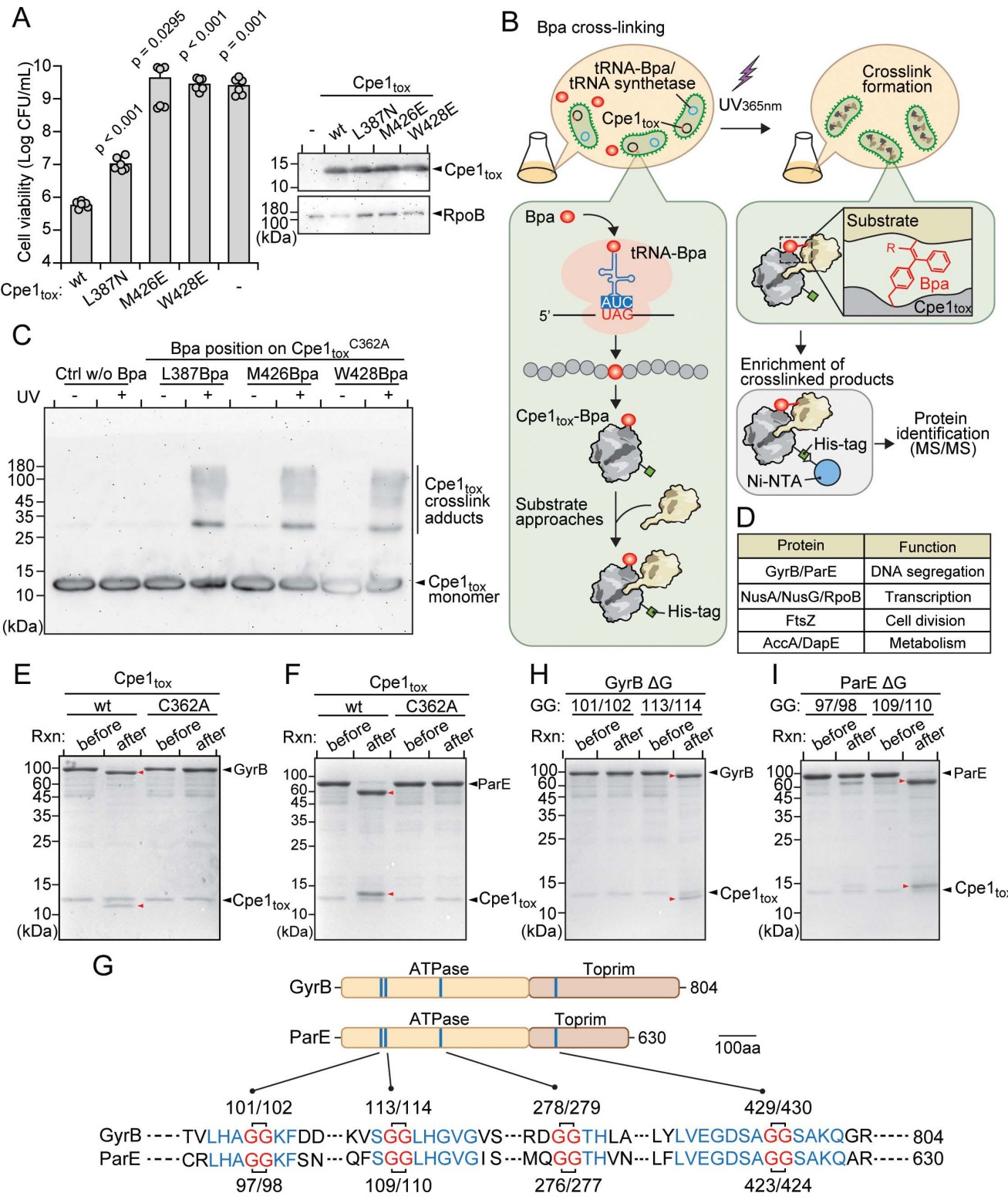

**Fig 3. Cpe1tox targets and cleaves the type II DNA topoisomerases GyrB and ParE. (a)** Viable *E. coli* cells recovered from expressing Cpe1tox with substitutions of indicated surface residues from the hydrophobic concave region. Data are shown as the mean ± SD; *n* = 6. The data shown are from a representative experiment of at least three independent experiments. *P*-values were calculated with Student *t* test to compare each population's viability

with that of cells expressing wild-type Cpe1$_{tox}$. The expression levels of wild-type and Cpe1$_{tox}$ variants were assessed by immunoblotting analysis, using the cytosolic protein RpoB as a loading control (right panel). **(b)** Schematic workflow for identifying candidate substrates of Cpe1$_{tox}$ through Bpa crosslinking followed by LC–MS/MS protein identification. Bacteria carrying the construct with a TAG stop codon at residues of interest were grown in medium supplemented with Bpa. After incubation, cells were exposed to long-wavelength UV to activate formation of covalent bonds between Bpa and the interacting residues. Cell lysates were incubated with Ni-NTA resin to enrich for Cpe1$_{tox}$-Bpa crosslinks for further analysis. **(c)** SDS–PAGE gel analysis of crosslinked protein complexes before and after UV$_{365nm}$ treatment. Cpe1$_{tox}$$^{C362A}$-protein complexes enriched by the Ni-NTA resin were boiled in SDS sample buffer and separated by SDS–PAGE, followed by immunoblotting (*a*-His). **(d)** Essential proteins identified in the Cpe1$_{tox}$$^{C362A}$ interactome and their biological functions. The complete interactome dataset is available in S3 Table. (e, f, h, **i)** *In vitro* Cpe1$_{tox}$ cleavage assays. Purified Cpe1$_{tox}$ (wild-type or catalytic mutant) were incubated with purified candidate substrates, *i.e.,* GyrB **(e)** and ParE **(f)**, or their variants with single-glycine deletion at the predicted cleavage sites (h, **i)**, to assess hydrolysis by Cpe1$_{tox}$. Fragmented proteins are marked with red arrowheads. Rxn means reaction. **(g)** Double glycine motifs in the consensus regions of *E. coli* GyrB and ParE. Locations of consensus regions between GyrB and ParE containing double glycine motifs are indicated on the cartoon representation of the full-length proteins. The sequences of the consensus regions are shown below, with the positions of the double glycine motifs labeled. The GG motif (red) and the surrounding residues (blue) are highlighted. The data underlying this figure can be found in S1 Data and S1 Raw Images. The raw proteome data can be found in https://doi.org/10.5281/zenodo.15361709.

toxicity (Fig 3a). We incorporated the photo-reactive amino acid p-benzoyl-L-phenylalanine (Bpa) in place of the selected amino acids using enhanced nonsense suppression [55] (Fig 3b). This crosslinking approach provides several advantages, including the ability to introduce the Bpa crosslinker at a single site within Cpe1, and its modestly larger size relative to phenylalanine increases the likelihood of capturing direct interactions with substrates [56]. To prevent Cpe1 from cleaving its bound substrates, we introduced the catalytically inactive C362A mutation into the Cpe1-Bpa variants for our crosslinking experiments.

Cells expressing these Cpe1-Bpa variants were directly exposed to long-wavelength UV irradiation (365 nm) to induce covalent crosslinking between Cpe1 and potential substrates (Fig 3b). Following UV exposure, Cpe1-Bpa and its crosslinked products were purified using the hexa-histidine tag located at the C-terminus of Cpe1 and analyzed by electrophoresis and immunoblotting. As expected, we observed multiple crosslinked products migrating more slowly by SDS–PAGE than Cpe1-Bpa monomers (*i.e.,* L387Bpa, M426Bpa, W428Bpa) (Fig 3c). The molecular weights of the crosslinked products ranged from 30 kDa to 180 kDa, indicating formation of Cpe1-substrate complexes. In contrast, our control samples, which did not incorporate Bpa, presented no high-molecular-weight crosslinked adducts on the gels. Additionally, levels of the Cpe1-Bpa monomers (L387Bpa, M426Bpa, W428Bpa) slightly decreased after UV irradiation, further supporting that Cpe1-Bpa bound and crosslinked to its substrates via the hydrophobic concave region. This decrease in level upon UV irradiation was not observed for a parallel control experiment, indicating that the crosslinking was specific to the modified Cpe1-Bpa variants (Fig 3c).

## Cpe1 targets the essential type II DNA topoisomerases GyrB and ParE

To define potential protein substrates of the Cpe1 toxin, we subjected purified Cpe1-Bpa crosslinked adducts to mass spectrometric analysis. To minimize false-positive results that could obscure our identification of physiological targets, we applied a stringent cutoff, focusing on proteins with >10 unique peptides, an intensity index > 5, and consistent detection across three replicates. Given that many interbacterial toxins associated with the T6SS and CDI pathways are known to target essential cellular components to inhibit or kill competing cells [48], we focused our analysis on essential proteins, though we are aware that non-essential proteins might also contribute to critical cellular processes.

Our mass spectrometry analysis revealed 51 cytosolic proteins that were consistently detected across all three Cpe1-Bpa crosslinking experiments (S3 Table). Among these, eight were identified as essential for *E. coli* viability, each playing a critical role in diverse cellular processes [57,58]. These proteins included the type II DNA topoisomerases GyrB and ParE, essential for DNA replication and chromosome segregation; NusA, NusG, and RpoB, involved in transcription; FtsZ, critical for cell division; and AccA and DapE, vital for metabolism (Fig 3d).

To validate the substrates of Cpe1, we performed an *in vitro* protease cleavage assay by incubating purified candidate target proteins with Cpe1$_{tox}$. In this assay, Cpe1$_{tox}$ is expected to recognize and cleave the targets, with proteolytic cleavage being confirmed by SDS–PAGE analysis, whereby a reduction in molecular weight and the presence of cleavage fragments indicated substrate digestion. As a result, we observed that two out of the eight essential proteins—type II DNA topoisomerases GyrB and ParE—displayed clear molecular weight reductions upon being incubated with Cpe1$_{tox}$ but not with the catalytically inactive Cpe1$_{tox}$$^{C362A}$ ([Figs 3e](), [3f]() and [S5a]()–[S5f]()). Moreover, adding the immunity protein Cpi1 to the reaction inhibited the proteolytic activity of Cpe1 ([S5g](), [S5h Fig]()). Notably, incubating either GyrB or ParE with wild-type Cpe1$_{tox}$ resulted in relatively short but consistent fragments (approximately 12–14 kDa), indicating that the cleavage events are specific and may occur toward the termini of these proteins. Collectively, these findings support a model in which Cpe1 targets the essential proteins GyrB and ParE, potentially impairing vital cellular functions as part of a competitive inhibition strategy.

## Cpe1 cleaves a consensus sequence containing double glycine motifs within GyrB and ParE

Many proteases in the PLCP family cleave peptide bonds at sequence-specific sites [51]. Notably, homologs of Cpe1, such as ComA and LahT, recognize and cleave their native substrates at sequences containing a double glycine (GG) motif [50,59,60] ([S6a](), [S6b Fig]()). Accordingly, we hypothesized that Cpe1 might also target its substrates, GyrB and ParE, at a consensus sequence containing two consecutive glycine residues. Protein sequence alignment of GyrB and ParE revealed four consensus sequences containing GG motifs ([Fig 3g]()). Two of these motifs, located near the N-terminal ATPase domain: $G_{101}G_{102}$ and $G_{113}G_{114}$ in GyrB, and $G_{97}G_{98}$ and $G_{109}G_{110}$ in ParE, were prioritized for further investigation, as cleavage at these sites would generate approximately 12–14 kDa fragments, *i.e.,* matching those observed in our *in vitro* cleavage assays ([Fig 3e](), [3f]()).

To confirm these cleavage sites, we purified variants of GyrB and ParE in which one glycine residue had been deleted from each GG motif, and then subjected them to Cpe1$_{tox}$ cleavage assays. Our results showed that removing one of the $G_{101}G_{102}$ residues in GyrB or from $G_{97}G_{98}$ in ParE significantly reduced Cpe1$_{tox}$-mediated cleavage ([Fig 3h](), [3i]()). In contrast, removing one glycine from the other GG motif in GyrB ($G_{113}G_{114}$) and ParE ($G_{109}G_{110}$) had no effect on Cpe1$_{tox}$ cleavage. To further characterize the cleavage sites of Cpe1 on its substrates, we analyzed Cpe1-cleaved GyrB and ParE fragments through in-gel digestion followed by mass spectrometry ([S6c](), [S6d Fig]()). LC-MS/MS analysis demonstrated that both GyrB and ParE were cleaved precisely between the glycine residues in the GG motifs located within their ATPase domains ([S6e](), [S6f Fig]()). Together, these findings support the notion that Cpe1 selectively targets consensus sequences containing the $G_{101}G_{102}$ motif in GyrB and the $G_{97}G_{98}$ motif in ParE, reinforcing the substrate specificity of Cpe1 ([Fig 3g]()).

## Cpe1-mediated proteolysis disrupts DNA topology and inhibits chromosome segregation in bacteria

GyrB and ParE are members of the type II DNA topoisomerases, forming the hetero-tetrameric DNA gyrase (GyrB$^2$/GyrA$^2$) and topoisomerase IV (ParE$^2$/ParC$^2$) complexes, respectively, in bacteria [61]. These enzymes are well characterized for their roles in catalyzing the relaxation of supercoiled DNA in an ATP-dependent manner, thereby facilitating critical biological processes that require modification of DNA topology [62]. Malfunction of DNA gyrase impairs DNA replication, transcription, and repair, whereas inhibition of topoisomerase IV results in abnormal catenation of daughter chromosomes following replication, ultimately blocking cell division [63]. These essential activities make both proteins appealing targets for antimicrobial agents in clinical settings [64,65]. Notably, the consensus sequence recognized by Cpe1 is located within a loop connecting helix 3 and helix 4 in the ATPase domain of GyrB and ParE [66, 67] ([Fig 4a](), [4b]()). This loop is highly conserved across different bacterial species [68] ([Fig 4c](), [4d]()), and has been previously identified as a critical region involved in ATP-binding, dimerization, and conformational changes triggered by ATP hydrolysis [67,69,70]. Therefore, cleavage of the loops in GyrB and ParE by Cpe1 would paralyze topoisomerase function, conferring a competitive fitness advantage to antagonistic bacteria by inhibiting DNA replication and chromosome segregation in their rivals.

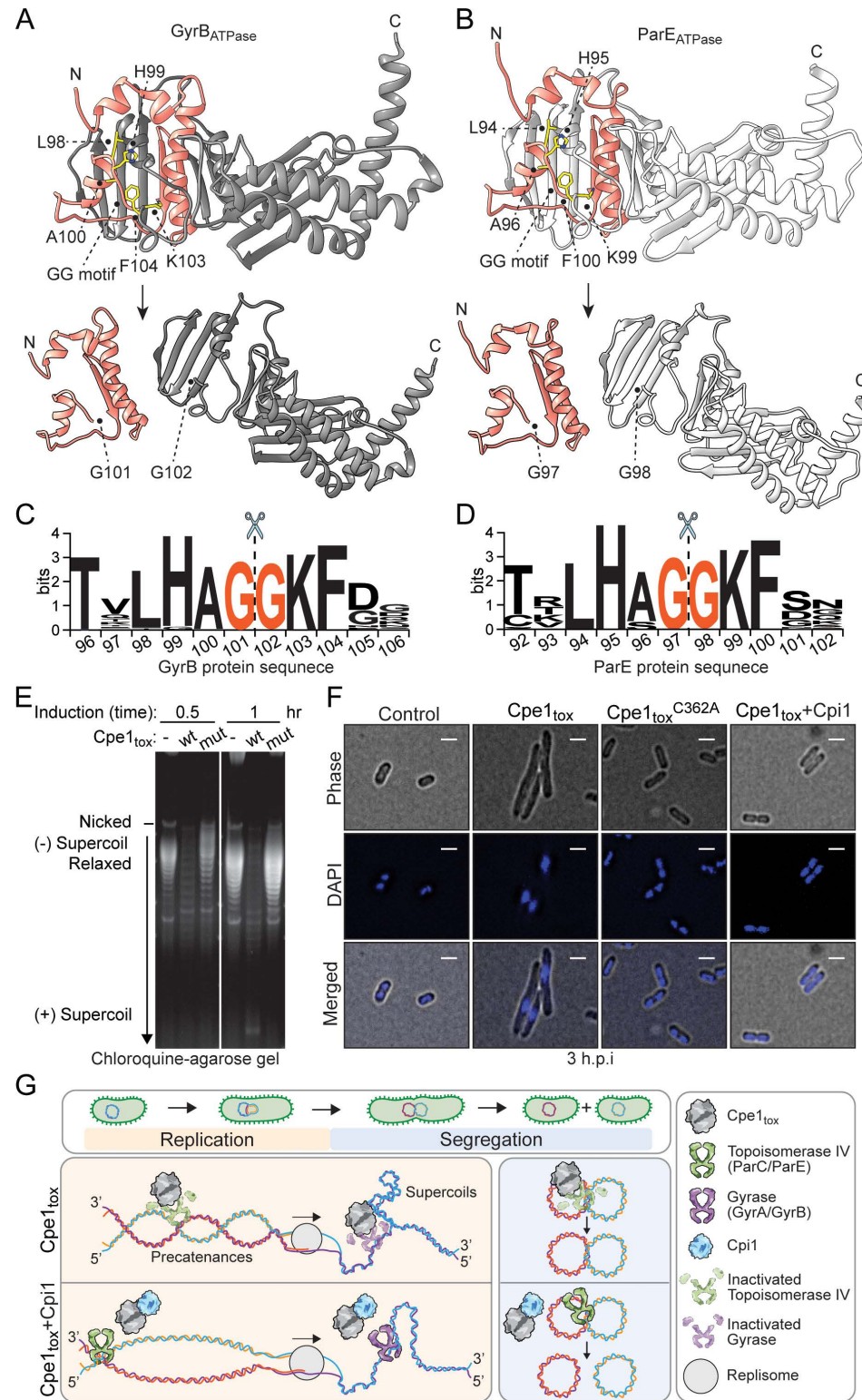

**Fig 4. Cpe1tox-mediated intoxication disrupts topoisomerase activity, thereby altering DNA topology and impairing DNA replication and chromosome segregation. (a–b)** Ribbon diagrams of *E. coli* GyrB (a, top structure) and ParE (b, top structure) before and after cleavage by Cpe1tox. Crystal structures of the 43-kDa ATPase domain of *E. coli* GyrB (PDB: 4PRV) and ParE (PDB: 1S16) were used. Residues of the consensus sequence

(LHAGGKF) are shown in yellow. Structure related to the small fragments after Cpe1$_{tox}$ cleavage are indicated as pink ribbons. Predicted structures after cleavage at the double glycine motif demonstrated disruption of the ATPase domain of both GyrB (a, bottom structure) and ParE (b, bottom structure). **(c, d)** Sequence logos evidencing conservation of the consensus sequence (LHAGGKF) of GyrB (c) and ParE (d) homologs. Cleavage at the double glycine motif is indicated by dashed lines and a scissors illustration. **(e)** Plasmid DNA (pET) was analyzed by high-resolution agarose gel electrophoresis following isolation from *E. coli* that had expressed Cpe1$_{tox}$ or the catalytic mutant for 0.5 or 1 h. The raw image is available in S1 Raw Images. **(f)** *E. coli* cells that had expressed Cpe1$_{tox}$, Cpe1$_{tox}$$^{C362A}$, or Cpe1$_{tox}$ co-expressed with Cpi1 for three hours. Cpe1$_{tox}$, or the catalytic mutant, was stained with DAPI (DNA; blue) and analyzed by fluorescence microscopy. Phase-contrast (top), blue fluorescence (middle), and merged (bottom) images are presented. Scale bar = 2 μm. Full-size images are available in S7 Fig. **(g)** Working model of how Cpe1 intoxicates bacteria. During interbacterial competition, Cpe1 targets and cleaves the ATPase domains of GyrB and ParE in target cells. The cleavage leads to accumulations of unrelaxed supercoils and precatenanes, disrupts DNA segregation, and ultimately inhibits cell division and growth. In the Cpe1-producing cells or kin cells expressing the immunity protein Cpi1, Cpe1 toxicity is neutralized through direct interaction between Cpe1 and Cpi1.

If GyrB and ParE are indeed substrates of Cpe1, we reasoned that Cpe1 intoxication would impair the ability of both enzymes to maintain DNA topology, specifically by decreasing negative supercoiling of cellular DNA. Therefore, we used a high-copy plasmid, pET, isolated from *E. coli* ectopically expressing Cpe1 to visualize changes in DNA topology via high-resolution agarose gel electrophoresis with chloroquine, allowing the resolution of negatively supercoiled DNA. As predicted, plasmid DNA topoisomers extracted from Cpe1-intoxicated cells (0.5 and 1 h of induction) displayed a significant decrease in negatively supercoiled DNA bands [71] (Fig 4e). In contrast, plasmid DNA from cells expressing the catalytic mutant Cpe1$^{C362A}$ showed a high level of plasmid DNA in a relaxed state, *i.e.,* similar to the vector control. This observation mirrors the effects reported previously when the ATPase domains of GyrB or ParE were inhibited or mutated, supporting that the observed changes were attributable to disruption of type II DNA topoisomerase activity [69,72,73].

Inactivation of topoisomerase IV is known to result in DNA knotting and catenation, leading to an accumulation of unsegregated DNA at the center of bacterial cells [74]. Fluorescence microscopy validated this notion, revealing unsegregated and condensed DNA in elongated bacteria subjected to Cpe1 intoxication (Figs 4f and S7), representing a phenotype reminiscent of cells treated with novobiocin, an antibiotic that inhibits the ATPase activities of both GyrB and ParE [75,76]. In contrast, sister nucleoids formed and segregated normally in dividing cells expressing either the Cpe1$^{C362A}$ variant or those co-expressing Cpe1 and its cognate immunity protein, Cpi1 (Figs 4f and S7). Given that chromosome partitioning is critical for successful cell division, next we examined if Cpe1 intoxication also disrupts the bacterial cell division machinery. To test this possibility, we conducted Cpe1 intoxication experiments on *E. coli* expressing fluorescently labeled cell division protein FtsZ (*i.e.,* FtsZ-mNeonGreen) [77]. As expected, the *Z*-ring structure that is essential for cell division failed to form at the septa of Cpe1-intoxicated bacteria (S8a, S8b Fig). These findings collectively support a model in which Cpe1 targets and cleaves the consensus loops in ATPase domains of GyrB and ParE, disrupting their topoisomerase activity and thereby inhibiting DNA replication, chromosome segregation, and cell division, ultimately resulting in the cessation of bacterial growth (Fig 4g).

## Cpi1 confers competitive inhibition against Cpe1

Our identification of GyrB and ParE as substrates of Cpe1 prompted us to investigate the mechanism by which Cpi1 inhibits Cpe1. Based on the Cpe1$_{tox}$–Cpi1 complex structure, toxicity assays, and site-specific crosslinking experiments, we suspected that Cpi1 binds competitively at the hydrophobic concave region of Cpe1, overlapping with the binding regions of GyrB and ParE, thereby neutralizing Cpe1-mediated intoxication. This competitive inhibition model was supported by AlphaFold3 (AF3) predictions, which revealed that the binding regions for GyrB and ParE on Cpe1 partially overlap with those of Cpi1 in the predicted structures [78] (Figs 5a, and S9a, S9b). Notably, the loop sequence containing the Cpe1 cleavage site on GyrB actually extends into the catalytic pocket of Cpe1 in the AF3 model structure, supporting our prior characterization of its protease activity and substrate specificity (S9c Fig).

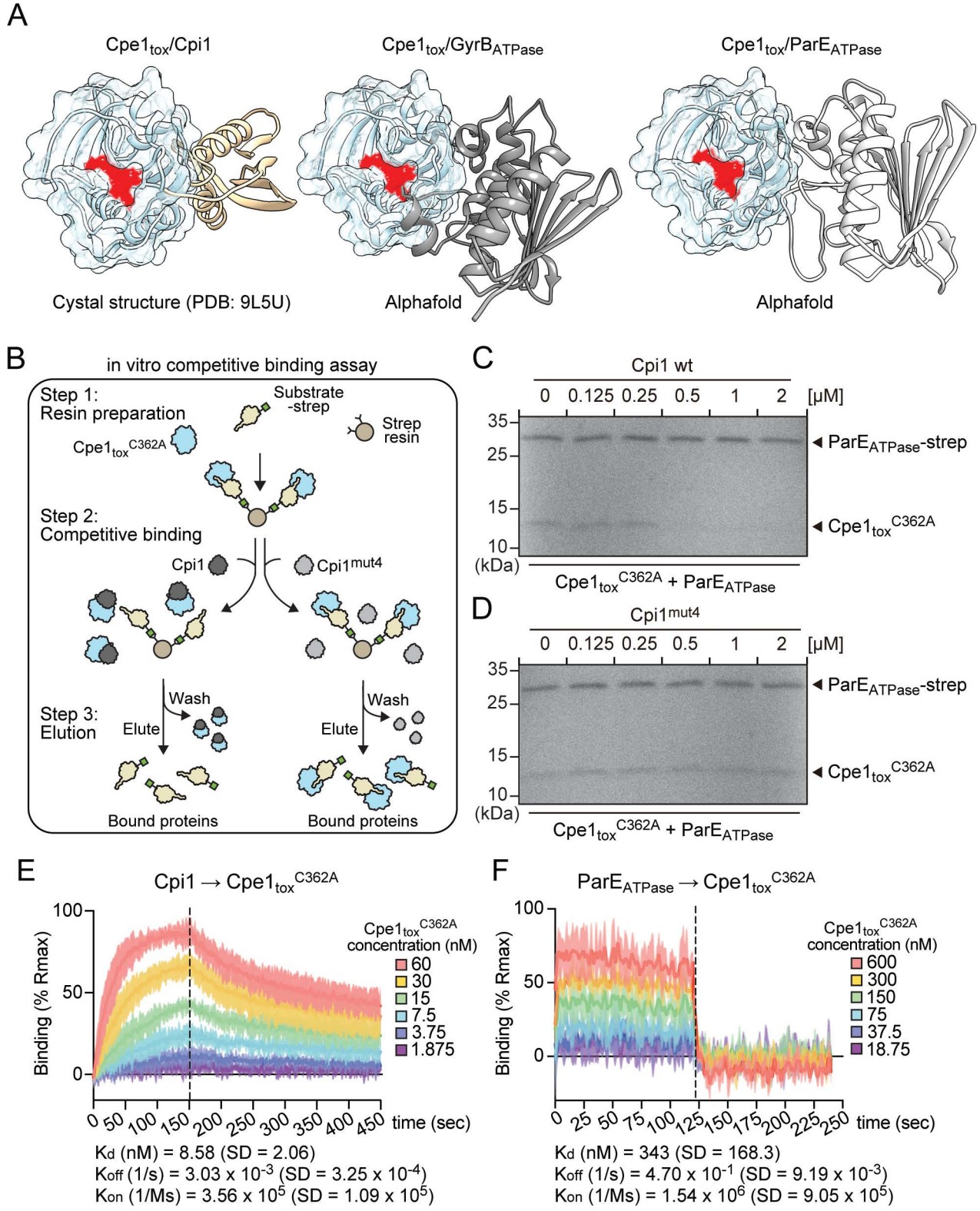

**Fig 5. Cpi1 employs a competitive inhibitory mechanism to neutralize Cpe1.** (a) The Cpe1$_{tox}$–Cpi1 interface overlaps with the Cpe1$_{tox}$-substrate interaction site. Model predictions of interaction between Cpe1$_{tox}$ and the ATPase domains of GyrB and ParE were constructed using AlphaFold 3. See

also <u>S9a</u>, <u>S9b</u> Fig for confidence metrics. **(b)** Schematic workflow for the *in vitro* competitive binding assay. Streptavidin resins bound to pre-formed Cpe1$_{tox}$$^{C362A}$-substrate complexes were prepared and incubated with either wild-type or mutant Cpi1. After incubation, the resins were washed, and the bound proteins were eluted for analysis. Competitive binding events are indicated by disruption of the Cpe1$_{tox}$$^{C362A}$-substrate interaction, leading to a decreased level of Cpe1$_{tox}$$^{C362A}$ in the eluted fraction due to its removal during the wash step. **(c, d)** Coomassie-stained SDS–PAGE demonstrating competitive binding events. **(e, f)** Kinetic analysis of interactions between Cpi1 and Cpe1$_{tox}$$^{C362A}$ (e) and ParE$_{ATPase}$ and Cpe1$_{tox}$$^{C362A}$ (f) using biolayer interferometry (BLI). Data from three replicates have been plotted on an *X–Y* scatter graph. The *Y*-axis represents the wavelength shift generated by the binding of the two proteins, normalized to the calculated $R_{max}$ (% $R_{max}$) for each run. The *X*-axis represents reaction time (in seconds). A vertical dashed line marks the transition from the association step to the dissociation step. The mean kinetic parameters ($K_d$, $K_{on}$, and $K_{off}$) and their standard deviations are indicated. The data underlying this figure are available in <u>S1 Data</u> and <u>S1 Raw Images</u>.

<u>https://doi.org/10.1371/journal.pbio.3003208.g005</u>

To test the competitive inhibition model, we began by conducting *in vitro* competitive binding assays to determine if Cpi1 could displace substrate interactions with Cpe1 (<u>Fig 5b</u>). In our assay, catalytically inactive Cpe1$^{C362A}$ was incubated with the substrate (*e.g.,* the ATPase domain of ParE), which was tagged with a C-terminal strep tag to allow complex formation. The Cpe1/ParE complex was then immobilized on streptavidin resin. The ATPase domain of ParE served as a representative substrate due to its ease of purification and susceptibility to cleavage by wild-type Cpe1, similar to full-length ParE (<u>S9d Fig</u>). Following complex formation, increasing concentrations of wild-type Cpi1 were added, followed by washing to remove unbound Cpe1$^{C362A}$. As a result, Cpe1$^{C362A}$ levels bound to ParE$_{ATPase}$ were markedly reduced in the presence of Cpi1 in a concentration-dependent manner, indicating that Cpi1 displaced ParE$_{ATPase}$ from its interaction with Cpe1$^{C362A}$ (<u>Fig 5c</u>). In contrast, in a control experiment using Cpi1$^{mut4}$, in which the Cpe1-binding residues had been mutated (<u>Fig 2j</u>), no significant changes in Cpe1$^{C362A}$ level were observed (<u>Fig 5d</u>). These findings support that Cpi1 competitively interacts with the substrate-binding site on Cpe1. Based on the observed competition, we reasoned that Cpi1 should exhibit a higher binding affinity to Cpe1 than its target substrates. Using biolayer interferometry, we confirmed that Cpi1 binds to Cpe1$^{C362A}$ with an approximately 40-fold higher affinity than ParE$_{ATPase}$ *in vitro* ($K_d$: Cpe1$^{C362A}$–Cpi1 = 8.58 nM; Cpe1$^{C362A}$–ParE$_{ATPase}$ = 343 nM). However, binding affinity was too low to be measured when Cpi1$^{mut4}$ was used (<u>Figs 5e</u>, <u>5f</u> and <u>S9e</u>). Collectively, these results support a model whereby Cpi1 neutralizes Cpe1-mediated intoxication by employing a competitive inhibition mechanism.

## Discussion

Our study characterizes a widely distributed class of antibacterial effectors displaying papain-like cysteine protease activity, referred to herein as Cpe1, which is found in diverse gram-negative bacteria. By targeting essential cellular processes in bacteria, these Cpe1 effectors play a critical role in bacterial competition, broadening the known repertoire of bacterial toxins involved in interbacterial antagonism. We have demonstrated that Cpe1 exerts its toxic effect by cleaving key protein substrates, specifically the type II DNA topoisomerases GyrB and ParE, which are indispensable for DNA replication and chromosome segregation. Cpe1 mediates this cleavage by targeting consensus sequences containing GG motifs within the ATPase domains of these proteins, thereby inactivating the topoisomerase function and halting the DNA replication machinery and cell growth. This mechanism provides a competitive advantage to Cpe1-producing strains.

Our phylogenetic analysis reveals a complex evolutionary trajectory for Cpe1 homologs, underscoring their occurrence across diverse bacterial phyla and their functional association with secretion systems such as the T6SS and CDI pathways (<u>Fig 1d</u>). The broad distribution of Cpe1 homologs indicates that horizontal gene transfer has facilitated adaptation of these protease effectors to diverse ecological niches, enabling bacteria to assert competitive dominance in various environments. Furthermore, the lack of correlation between phylogeny and pathway association among Cpe1 homologs supports that selective pressures favor the adoption of Cpe1-like effectors across multiple bacterial lineages, thereby enhancing bacterial fitness in different ecological contexts.

Papain-like cysteine proteases are versatile enzymes that cleave peptide bonds in protein targets and are frequently employed as toxic effectors by bacterial pathogens. For example, the bacterial effector OspD3, secreted by *Shigella flexneri*, cleaves host kinases RIPK1 and RIPK3 to evade necroptosis-mediated immune responses during infection [39]. Similarly, the YopT effector from *Yersinia sp.* cleaves Rho-family GTPases, such as RhoA, leading to disruption of the actin cytoskeleton [79]. Additionally, AvrPphB from *Pseudomonas syringae*, induces plant cell death by cleaving the PBS1 kinase through its cysteine protease activity [80,81]. Our findings provide compelling evidence of bacteria leveraging a PLCP-mediated mechanism to intoxicate competing bacteria, as demonstrated by Cpe1 cleavage of essential DNA topoisomerases. Given that interbacterial interactions predate the evolution of eukaryotic cells, it is plausible that PLCP toxins targeting eukaryotic hosts evolved from ancestral PLCPs possessing antibacterial activity.

We identified several cellular proteins crosslinked to Cpe1-Bpa from our crosslinking mass spectrometry experiments (Fig 3b–d). In addition to GyrB and ParE, six other essential proteins were detected, namely NusA, NusG, RpoB, FtsZ, AccA, and DapE. However, Cpe1 exhibited cleavage activity toward GyrB and ParE in our *in vitro* protease cleavage assays, but not against these other six targets (Figs 3e, 3f and S5a–S5f). These false- positive identifications might be due to non-specific interactions with heterologous Cpe1-Bpa expression. Our further analysis of the cleavage sites revealed that Cpe1 recognizes a consensus sequence, LHAGGKF, within a loop region of its substrates and cleaves the peptide bond immediately following the double glycine residues. Deletion of these glycine residues in GyrB and ParE significantly reduced Cpe1 activity *in vitro* (Fig 3h, 3i). The contribution of other amino acid residues within the consensus sequence to Cpe1 activity remains unclear, as does the influence of substrate three-dimensional structure on Cpe1 specificity and cleavage. Systematic analysis of cleavage sequences and structural determinants is required to determine if Cpe1 can recognize and cleave additional substrates in bacteria.

During the manuscript revision period, we attempted to overexpress the cleavage-resistant variants GyrB$^{\Delta G102}$ and ParE$^{\Delta G98}$ in anticipation that they might confer resistance to Cpe1 intoxication. However, despite robust overexpression of these variants (S10a, S10b Fig), they did not show significant protective effects against Cpe1-mediated killing in either the co-expression or interbacterial competition assays (S10c, S10d Fig). This finding suggests that cleavage of GyrB and ParE may not be the sole mechanism underlying Cpe1 toxicity. While no additional proteins in *E. coli* were found to contain the full LHAGGKF motif, it remains plausible that Cpe1 targets other substrates with partial sequence similarity or compatible structural features. Alternatively, non-mutually exclusive explanations may also account for the lack of protection conferred by the cleavage-resistant mutants. In the co-expression assay, high levels of Cpe1 may continue to act on endogenous wild-type GyrB and ParE, outcompeting the mutant forms despite their overexpression. Similarly, in the competition assay, ongoing cleavage of wild-type substrates throughout the experiment may obscure any protective effect. Moreover, the activity or proper folding of the mutant proteins may be partially compromised and contribute to the observed results. Finally, we note that deletion of the global transcriptional repressor H-NS in the Cpe1-producing strain may have activated other interbacterial antagonism pathways, contributing to growth inhibition independently of Cpe1 [82]. Collectively, these findings highlight the complexity of interbacterial intoxication systems and underscore the need for further investigation to elucidate the full repertoire of Cpe1 substrates and mechanisms of action.

Our results demonstrate that Cpe1 specifically targets the ATPase domains of the type II DNA topoisomerases GyrB and ParE, disrupting their essential roles in maintaining DNA topology and chromosome segregation, ultimately leading to cell division arrest. The coupling of ATP hydrolysis to the strand-passage process is a crucial step in the supercoiling reaction of type II DNA topoisomerases, making the ATP-binding site a well-established target for enzyme inhibition. For instance, the FicT toxin, which exhibits adenylyl transferase activity, inactivates GyrB and ParE by post-translationally modifying conserved residues (Tyrosine 109 in GyrB and Tyrosine 105 in ParE), thereby abolishing ATP hydrolysis and arresting bacterial growth through impaired DNA topology regulation [83,84]. Interestingly, we observed similar defects in DNA topology and segregation in Cpe1-intoxicated bacteria (Fig 4e, 4f), likely as a result of ATPase dysfunction. Notably, our study reveals that Cpe1 induces irreversible cleavage at a conserved loop within the ATPase domains of GyrB and

ParE (Fig 4a, 4b), distinguishing its mechanism from the reversible inhibition observed for the Fic toxin. These results suggest that conserved regions within the ATPase domains of type II DNA topoisomerases represent potential hotspots for the design of novel inhibitors and highlight the potential of Cpe1 as an irreversible topoisomerase inhibitor.

In addition to elucidating the role of Cpe1 during interbacterial antagonism, our study identifies a unique immunity mechanism that safeguards Cpe1-producing bacteria from self-toxicity. The cognate immunity protein, Cpi1, binds to a conserved hydrophobic concave surface adjacent to the catalytic pocket of Cpe1 (Fig 2f). This interaction prevents Cpe1 from accessing its substrates without occluding the active site of the effector. The coevolutionary relationship between Cpe1 and Cpi1 is evidenced by sequence conservation and structural complementarity. The evolutionary advantage of this mode of inhibition likely lies in the enhanced binding affinity afforded by the large surface area of hydrophobic interaction. Similar mechanisms have been reported for other protease-inhibitor systems. For example, the anticoagulation factor triabin inhibits thrombin protease by binding to regions outside of the catalytic pocket [85,86], and calpastatin neutralizes calpain protease through an exosite interaction, tripling the binding surface compared to other inhibitors [87,88]. Likewise, the anti-toxins of colicin, such as E3 rRNase and E9 DNase, are known to neutralize these toxins by binding with high affinity to the exosites, effectively blocking activity without overlapping the active site [89].

Bacterial toxins possessing protease activity have long been proven valuable tools in investigating host cell biology due to their ability to manipulate with precision specific cellular functions. For example, the identification of the protein targeted by the specific endopeptidase activity of the tetanus and botulinum toxins provided fundamental insights into the role of SNARE complex proteins in regulating exocytosis [90,91]. Our finding that Cpe1 specifically cleaves GyrB and ParE has unveiled the possibility of applying this toxin to study DNA replication, catenation, and chromosome segregation, among other aspects of cell biology. Although the general functions of these topoisomerases are already well established, certain mechanistic aspects remain unclear [92]. Given its precise targeting of the ATPase domains of GyrB and ParE, Cpe1 could serve as a valuable tool for dissecting the finer details of certain critical processes.

In conclusion, our study provides a detailed characterization of Cpe1 as a potent interbacterial toxin that targets essential DNA topoisomerases in competitor cells. By broadening the functional scope of PLCPs in bacterial competition and elucidating the molecular mechanisms underlying immunity against Cpe1, our findings lay the groundwork for further exploration of how PLCPs function within microbial communities. The substrate specificity and potent activity of Cpe1 highlight its potential for applications in developing analogs or inhibitors that replicate its targeting mechanism. Furthermore, insights into the Cpi1 immunity mechanism could inform the engineering of bacterial strains presenting enhanced resistance to PLCP-mediated antagonism, opening up avenues for biotechnological applications such as microbiome manipulation and the creation of next-generation antibacterial therapies.

## Methods

### Bioinformatic analysis of Cpe1 and Cpi1-related proteins

The protein sequences of the full-length PLCP effectors EvpQ (GenBank: JACQ01000015.1, 119,565−120,092 basepairs) and TsaP (accession number: AJM72455.1) were obtained from the NCBI database, as described in previous studies [36,37]. The cysteine protease domains within these sequences were identified using the NCBI Conserved Domain Search tool [93] and subsequently used as seed sequences for PSI–BLAST [45]. PSI–BLAST was run iteratively until convergence, with a relatively lenient e-value cut-off of 0.01 to ensure inclusion of highly divergent homologs. To minimize false-positive hits, the retrieved sequences were filtered to retain only those with a minimum length of 100 amino acids.

Interbacterial PLCP effectors were manually curated based on their genomic context within gene clusters associated with contact-dependent antagonistic pathways (*i.e.,* T6SS, CDI) [94,95]. Effector-associated domains known to mediate interbacterial antagonism, such as PAAR, VgrG, and Rhs for T6SS, or VENN and filamentous hemagglutinin peptide repeats for CDI, were used to validate their functional relevance [46,47]. Taxonomic and strain-level information for the identified PLCP effectors was retrieved from NCBI. Multiple sequence alignments of the homologs were performed using

ClustalW [96], and a maximum likelihood phylogenetic tree was constructed using the IQtree online tool with default parameters [97]. The resulting tree was visualized and edited using figtree (http://tree.bio.ed.ac.uk/software/figtree/).

### Bacteria and culture conditions

The bacterial strains used in this study are listed in S4 Table. The following *Escherichia coli* strains were used: DH5*α* and PIR1 for cloning and plasmid maintenance; S17-1 λ pir for conjugal transfer of the pRE118 plasmid into *E. coli*; and BL21 (DE3) for overproduction and purification of protein samples. *E. coli* ATCC-11775 and MG1,655 were used for toxicity assay and bacterial competition assays. Bacterial strains were routinely grown in Luria Bertani (LB) medium supplemented with appropriate antibiotics at 37 °C with shaking at 200 rpm. Antibiotics and chemicals were used at the following concentrations: 50 μg/mL for kanamycin, 25 μg/mL for chloramphenicol, 150 μg/mL for carbenicillin, 0.25 mM isopropyl β-D-1-thiogalactopyranoside (IPTG), 0.2% (*w/v*) L-(+)-arabinose, and 10% (*w/v*) sucrose.

### Plasmid construction

All plasmids and primers used in this study are listed in S4 Table. Mutations in *E. coli* were generated using the pRE118 vector [98]. For chromosomal mutation constructs, 800-basepair regions flanking the mutation site were amplified by polymerase chain reaction (PCR), sewn together, and inserted into the vector using Gibson assembly at the *XbaI* and *SacI* restriction sites [99]. Insertions at the Tn7 sites of *E. coli* strains were generated using the pUC18mini R6K vector with the target genes amplified by PCR and inserted into the vector through Gibson assembly at the *XhoI* and *SacI* restriction sites [100].

For co-expression and purification of the Cpe1$_{tox}$–Cpi1 complex, the *cpe1*$_{tox}$ and *cpi1* genes were amplified and cloned into MCS (multiple cloning site)-1 (*BamH1* and *NotI* sites) and MCS-2 (*NdeI* and *XhoI* sites) of the pETDuet-1 vector, respectively, resulting in N-terminal hexa-histidine fusion to *cpe1*$_{tox}$. Candidate protein substrates of Cpe1$_{tox}$ were purified by cloning the corresponding genes into MCS-1 of pETDuet-1. The *BamHI* and *NotI* sites were used to N-terminally fuse hexa-histidine to substrates (*e.g.,* GyrB and ParE) [101]. Alternatively, the *NcoI* and *NotI* sites were employed for C-terminal hexa-histidine fusions to substrates (*e.g.,* NusA, NusG, RpoB, AccA, and DapE) or for expressing proteins without fusion tags (*e.g.,* FtsZ). For strep-tagged proteins (*e.g.,* Cpi1 variants and the ATPase domain of ParE) and His-tagged Cpi1, the corresponding genes were cloned into MCS-2 of pETDuet-1 using the *NdeI* and *XhoI* restriction sites.

Site-directed mutagenesis was performed to generate substitution mutants of toxin and immunity proteins in pBAD33 and pETDuet-1 plasmids. Primers containing the desired mutations were paired with a complementary primer in the opposite orientation to amplify a mutated copy of the plasmid. The template plasmid was digested using *DpnI*, and the resulting mutated amplicon was transformed into *E. coli* competent cells. Mutant plasmids were selected by growth on plates containing carbenicillin (for pETDuet-1) or chloramphenicol (for pBAD33).

### Generation of mutant strains of *E. coli*

To generate mutations of *E. coli* ATCC-11775 strains, mutant constructs in pRE118 plasmid were transformed into *E. coli* S17-1 λ pir. *E. coli* S17-1 λ pir donor strains carrying the mutant constructs and *E. coli* ATCC-11775 recipient strains to be mutated were grown overnight on LB plates containing appropriate antibiotics, then scraped together to create a 1:1 mixture of each donor-recipient pair that was spread on an LB agar plate and incubated at 37 °C for 6 h to facilitate plasmid transfer via conjugation. Cell mixtures were then scraped into LB and plated on LB agar containing kanamycin and chloramphenicol to select for *E. coli* ATCC-11775 with integrated plasmid. *E. coli* ATCC-11775 merodiploid strains were then grown overnight in a non-selective LB medium at 37 °C, followed by counter-selection by plating on nutrient agar plates supplemented with sucrose. Kanamycin-sensitive, sucrose-resistant colonies were screened for allelic replacement by colony PCR, and mutations were confirmed by Sanger sequencing of the PCR products.

To generate *E. coli* MG1,655 strains with insertions at the Tn7 site, pUC18mini_R6K plasmids carrying the insert fragments and the helper plasmid pTNS3 were co-transformed into *E. coli* MG1655 through electroporation [100]. Following electroporation, the cell suspension was incubated in SOB media at 37 °C for 1 h for recovery. The cells were then plated onto LB agar containing kanamycin for selection. Kanamycin-resistant colonies were screened for the presence of insertions using colony PCR, and mutations were confirmed by Sanger sequencing of the PCR products.

### Cpe1 toxicity assays

To assess the toxicity of $Cpe1_{tox}$ variants and the protective effect conferred by expression of Cpi1 variants, overnight cultures of *E. coli* BL21 (DE3) harboring toxin and immunity genes in pBAD33 and pETDuet-1 vectors, respectively, were used to inoculate LB medium. Cultures were adjusted to an initial optical density ($OD_{600}$) of 0.05 and incubated at 37 °C with shaking at 200 rpm for 4 h in the presence of appropriate inducers. Following incubation, 10-fold serial dilutions of each culture were prepared and plated onto LB agar to determine colony-forming units (CFUs). Unless otherwise specified, $Cpe1_{tox}$ expression was induced by the addition of arabinose to cultures harboring pBAD33, whereas expression of Cpi1 from pETDuet-1 was induced by the addition of IPTG.

### Bacterial competition assays

For each competition experiment, *E. coli* ATCC-11775 Δ*hns* was used as the donor strain, whereas *E. coli* MG1,655 served as the recipient strain. The Δ*hns* mutation was introduced into the donor strain to up-regulate T6SS activity as previously described [102–104]. The indicated *E. coli* donor and recipient strains were cultured for 12 h in LB broth at 37 °C. The overnight cultures were diluted in LB medium to an $OD_{600}$ of 0.2 and incubated at 37 °C for 2 h. After incubation, cells were harvested by centrifugation at 2,300 × *g* for 5 min. The supernatant was discarded, and the cell pellets were washed once with LB medium. Donor and recipient strains were then mixed at a 50:1 $OD_{600}$ ratio, and the initial donor-to-recipient ratios were determined by performing 10-fold serial dilutions and plating on selective media. Competitions were initiated by spotting 3 × 5 μL of each mixture onto nitrocellulose filters placed on LB agar plates containing 3% (*w/v*) agar. The plates were incubated at 37 °C for 4 h. Following incubation, cells were harvested by scraping individual spots from excised sections of the nitrocellulose filter into LB medium. The suspensions were serially diluted and plated on selective media to quantify CFUs. The competitive index was calculated as the ratio of the output donor-to-recipient ratio divided by the input donor-to-recipient ratio.

### Analysis of protein expression levels

To analyze the expression of $Cpe1_{tox}$ and Cpi1 variants, *E. coli* strains harboring the plasmids expressing the target proteins were cultured in LB medium supplemented with appropriate inducer at 37 °C. Cells were harvested at an $OD_{600}$ of 1, and cell pellets were resuspended in SDS sample buffer for analysis. The samples were boiled at 100 °C for 10 min, and then subjected to SDS–PAGE. Proteins were transferred onto nitrocellulose membranes, which were then blocked in TBST buffer (10 mM Tris-HCl pH 7.5, 150 mM NaCl, 0.1% (*v/v*) Tween-20) containing 5% (*w/v*) non-fat milk for 30 min at room temperature. Blocked membranes were incubated with primary antibodies (*α*-His, *α*-Strep, or *α*-RNA Polymerase β) diluted in TBST for 1 h at room temperature. Following incubation, the membranes were washed with TBST and subsequently incubated for 30 min at room temperature with a secondary antibody (*α*-mouse IgG HRP-conjugated) diluted 1:5000 in TBST. After the wash with TBST, protein detection and visualization were performed using a UVP BioSpectrum 815 system.

### Protein purification

To prevent cytotoxicity during overexpression in *E. coli*, all $Cpe1_{tox}$ constructs were co-expressed with Cpi1 from pETDuet-1 in *E. coli* BL21 (DE3). Stationary-phase cultures of the expression strains were used to inoculate 1 l of LB

broth supplemented with carbenicillin, which was then incubated in a shaking (200 rpm) incubator at 37 °C until the mid-log phase ($OD_{600}$ = 0.4–0.7). Protein expression was induced by adding IPTG, followed by incubation at 30 °C for 5 h. Cells were harvested by centrifugation at 7,000 × $g$ for 30 min and resuspended in lysis buffer (50 mM Tris-HCl pH 7.5, 500 mM NaCl, 5 mM imidazole, 1 mM DTT, 5% ($w/v$) glycerol, and 1 mg/mL lysozyme). Resuspended cells were incubated on ice for 30 min, disrupted by sonication (10 pulses of 10 s each), and centrifuged at 40,000 × $g$ for 30 min to remove cellular debris. His-tagged $Cpe1_{tox}$ and associated proteins were purified from the supernatant using a 2 mL Ni-NTA agarose column by gravity flow. Bound proteins were eluted using a linear imidazole gradient, reaching a final concentration of 300 mM. Purity was assessed by SDS–PAGE, followed by Coomassie Brilliant Blue staining. For further purification, protein samples were subjected to fast protein liquid chromatography (FPLC) using gel filtration on a HiLoad 16/600 Superdex 200 pg column (GE Healthcare). Fractions demonstrating high purity were collected and used in biochemical assays. For selenomethionine-incorporated $Cpe1_{tox}$–Cpi1 complex, cells were grown in SelenoMethionine Medium Complete (Molecular Dimensions) under the same expression conditions described above. Cell lysis, Ni-NTA affinity purification, and gel filtration were performed as detailed for native proteins.

To obtain free $Cpe1_{tox}$ for experiments, a denaturation and refolding strategy was employed to separate His-tagged $Cpe1_{tox}$ from Cpi1. The protein mixture was first diluted 1:25 into denaturation buffer (20 mM Tris-HCl pH 7.5, 300 mM NaCl, 1 mM DTT, and 8 M urea) to denature the protein complex. Denatured proteins were then loaded onto a column containing 1 mL of Ni-NTA agarose, allowing selective binding of His-tagged $Cpe1_{tox}$ while dissociated Cpi1 was removed in the flow-through. On-column refolding was achieved by sequential washing with renaturation buffer (20 mM Tris-HCl pH 7.5, 300 mM NaCl, and 1 mM DTT). The refolded protein was subsequently eluted with elution buffer (20 mM Tris-HCl pH 7.5, 300 mM NaCl, and 300 mM imidazole). Eluted protein samples were dialyzed against storage buffer (20 mM Tris-HCl pH 7.5, 300 mM NaCl, 1 mM DTT, and 5% ($w/v$) glycerol) to remove residual imidazole and stabilize the protein. Dialyzed samples were then aliquoted and stored at −80 °C for subsequent experiments.

To purify Cpe1 candidate protein substrates (GyrB, ParE, NusA, NusG, RpoB, AccA, and DapE) and Cpi1, His-tagged proteins were expressed in $E.\ coli$ BL21 (DE3) transformed with the pETDuet-1 expression vector containing the corresponding coding regions. Cultures were grown in LB broth at 37 °C to mid-log phase and induced with IPTG, followed by 2-h incubation at 37 °C. Cells were harvested by centrifugation, resuspended in lysis buffer (20 mM Tris-HCl pH 7, 300 mM NaCl, 10 mM imidazole, 1 mM DTT, and 1 mg/mL lysozyme), and incubated on ice for 30 min. The cells were disrupted by sonication (10 pulses, 10 s each), and the lysate was cleared by centrifugation at 40,000 × $g$ for 30 min. His-tagged proteins were purified from the supernatant by means of gravity flow through a 1 mL Ni-NTA agarose column. Proteins bound to the column were eluted with a linear imidazole gradient up to 300 mM. The purity of each protein sample was verified by SDS–PAGE followed by Coomassie Brilliant Blue staining. Purified proteins were resuspended and dialyzed into storage buffer (20 mM Tris-HCl pH 7, 300 mM NaCl, 1 mM DTT, and 5% ($w/v$) glycerol). Protein aliquots were frozen and stored at −80 °C for further use.

For purification of FtsZ, FtsZ was overproduced in $E.\ coli$ BL21 (DE3) transformed with the pETDuet-1 expression vector containing the FtsZ coding region. Cultures were grown in LB medium at 37 °C to mid-log phase and induced with IPTG, followed by 3-h induction at 37 °C. The cells were harvested by centrifugation, resuspended in lysis buffer (50 mM Tris-HCl pH 8, 100 mM NaCl, 1 mM EDTA, 5 mM $MgCl_2$, 1 mM PMSF, and 0.1 mg/mL lysozyme), and incubated on ice for 30 min. Cell lysis was performed by sonication (10 pulses, 10 s each), and the lysate was cleared by centrifugation at 40,000 × $g$ for 30 min. To remove unwanted proteins, the supernatant was subjected to ammonium sulfate fractionation. Proteins precipitated at 20% ($w/v$) ammonium sulfate were discarded, whereas those precipitated at 30% ($w/v$) ammonium sulfate were collected by centrifugation at 40,000 × $g$ for 30 min. The resulting pellet, enriched for FtsZ, was resuspended in buffer containing 20 mM Tris-HCl pH 8, 100 mM NaCl, 1 mM EDTA, and 5 mM $MgCl_2$. The resuspended proteins were dialyzed against storage buffer (20 mM Tris-HCl pH 8, 100 mM NaCl, 0.1 mM EDTA, and 5% ($w/v$) glycerol), aliquoted, and stored in a freezer at −80 °C.

For purification of proteins with strep tags (*i.e.,* Cpi1 variants and the ATPase domain of ParE), the proteins were over-produced in *E. coli* BL21 (DE3) transformed with the pETDuet-1 expression vector containing the respective gene coding regions. Cultures were grown in LB medium at 37 °C to mid-log phase and induced by the addition of IPTG. After 2-h induction at 37 °C, the cells were harvested by centrifugation and resuspended in lysis buffer (20 mM Tris-HCl pH 7, 300 mM NaCl, 1 mM DTT, and 1 mg/mL lysozyme). The cells were incubated on ice for 30 min, followed by sonication (10 pulses, 10 s each) to achieve lysis. Cellular debris was removed by centrifugation at 40,000 × $g$ for 30 min. Strep-tagged proteins were purified from the supernatant using a 1 mL gravity-flow column containing streptavidin resin (IBA Lifesciences GmbH). Bound proteins were eluted with a buffer containing 20 mM Tris-HCl pH 7, 300 mM NaCl, 1 mM DTT, and 2.5 mM desthiobiotin. Protein purity was evaluated by SDS–PAGE, followed by Coomassie Brilliant Blue staining. Samples were further purified by means of FPLC on a HiLoad 16/600 Superdex 200 pg column (GE Healthcare).

## Crystallization and structure determination

The Cpe1$_{tox}$–Cpi1 complex was prepared for crystallization by dialyzing the purified protein complex into a buffer containing 5 mM Tris-HCl (pH 7), 150 mM NaCl, and 0.5 mM tris(2-carboxyethyl)phosphine (TCEP). The protein solution was then concentrated to 13.5 mg/mL using spin filtration (10 kDa cutoff, Millipore). Crystallization trials were conducted using commercially available screens from Hampton Research. Diffraction-quality crystals were obtained after two weeks at 20 °C using the hanging-drop vapor diffusion method with a protein-to-buffer ratio of 1:2. The crystallization solution consisted of 0.4 M magnesium formate and 0.1 M sodium acetate (pH 4.6). Crystals of the selenomethionine-labeled Cpe1$_{tox}$–Cpi1 complex were grown under identical conditions as the native complex, with the protein sample also concentrated to 13.5 mg/mL and mixed at a 1:2 ratio with the same crystallization solution. Crystals were cryo-protected using a solution of 20% (*v/v*) glycerol in the crystallization mother liquor prior to data collection.

Native and multiple-wavelength anomalous diffraction (MAD) datasets were collected at beamlines TPS-05A and TLS-BL15A of the National Synchrotron Radiation Research Center (Taiwan). Diffraction data were processed using the HKL2000 software suite [105]. Phase determination was performed using PHENIX with data collected at the selenium peak wavelength from selenomethionine-labeled crystals [106]. The initial model of the Cpe1$_{tox}$–Cpi1 complex was constructed using the electron density map in Coot and subsequently refined using PHENIX [106,107]. The crystals belonged to the cubic space group P43212, with two protein complexes per asymmetric unit. No residues were found in the disallowed regions of the Ramachandran plot. Data collection and refinement statistics are provided in S2 Table.

## Bpa photo-crosslinking experiment

*E. coli* BL21 (DE3) cells were transformed with plasmids encoding Cpe1$_{tox}$$^{C362A}$ containing a TAG stop codon at specific sites (*i.e.,* L387, M426, and W428) to incorporate Bpa via nonsense suppression during translation. The transformation also included the Bpa-tRNA plasmid (pSUPT/BpF), which encodes the tRNA and tRNA synthetase necessary for Bpa incorporation, as described previously [108].

The transformed cells were grown in LB medium supplemented with 1.5 mM of Bpa at 37 °C until reaching the mid-log phase. Protein expression was induced by adding IPTG and arabinose, and the cultures were incubated at 30 °C for 3 h. Cells corresponding to 50 OD$_{600}$ units were harvested, resuspended in 1 mL of ice-cold phosphate-buffered saline (PBS), and divided into two aliquots. One aliquot was exposed to 365 nm UV light (Stratalinker 1800 UV irradiator) for 30 min at 4 °C to induce crosslinking, while the other aliquot was kept on ice as a non-crosslinked control. Cell lysis was performed by sonication in lysis buffer (20 mM Tris-HCl pH 7.5, 300 mM NaCl, 10 mM imidazole, 8 M urea, and 1 mM DTT). The lysates were incubated with Ni-NTA resin for 2 h at 4 °C to purify His-tagged proteins. Bound proteins, including cross-linked products, were analyzed using immunoblotting and mass spectrometry.

## Mass spectrometry

For our interactome analysis, Bpa-crosslinked Cpe1$_{tox}$$^{C362A}$-protein complexes were purified and immobilized on Ni-NTA resins. After washing four times in lysis buffer, the resins were resuspended in 8 M urea and Tris-HCl pH 8. For cysteine alkylation, 10 mM TCEP and 40 mM 2-chloroacetamide (CAA) were added into the samples and incubated at 45 °C for 30 min with shaking (1,500 rpm). The samples were diluted 4-fold with 50 mM Tris-HCl pH 8 and digested by 0.4 µg Lys-C (Wako, Japan) treatment for 3 h at room temperature and 1 µg of trypsin (Sigma, MO) overnight at 37 °C. Supernatants containing digested peptides were acidified by 1% TFA, desalted using the Oasis HLB 1 cc Vac Cartridge (Waters Corp), reconstituted in 80% acetonitrile (ACN), and vacuum-dried at 45 °C.

NanoLC−nanoESi−MS/MS analysis was performed on a Thermo UltiMate 3,000 RSLCnano system connected to a Thermo Orbitrap Fusion mass spectrometer (Thermo Fisher Scientific, Bremen, Germany) equipped with a nanospray interface (New Objective, Woburn, MA). Peptide mixtures were loaded onto a 75 µm ID, 25 cm long PepMap C18 column (Thermo Fisher Scientific) packed with 2 µm particles with a pore size of 100 Å and they were separated using a segmented gradient from 5% to 25% solvent B (0.1% formic acid in ACN) in 82.5 min and to 35% solvent B in 7.5 min at a flow rate of 300 nL/minute. Solvent A was 0.1% formic acid in water. The mass spectrometer was operated in the data-dependent mode. In brief, survey scans of peptide precursors from 350 to 1,600 m/z were performed at 120 K resolution with a $2 \times 10^5$ ion count target. Tandem MS was performed by isolation window at 1.6 daltons with the quadrupole, HCD fragmentation with a normalized collision energy of 30, and MS 2 scan analysis at 30 K resolution in the orbitrap. The MS 2 ion count target was set to $5 \times 10^4$ and the max injection time was 54 ms. Only those precursors with charge states 2–6 were sampled for MS 2. The instrument was run in top speed mode with 3-s cycles. The dynamic exclusion duration was set to 11 s with a 10-ppm tolerance around the selected precursor and its isotopes. Monoisotopic precursor selection was turned on. Raw data were processed using MaxQuant software [109,110].

To identify digested peptides of Cpe1, the samples were prepared based on a previous study with minor modifications [111]. The protein band from SDS–PAGE was manually excised and cut into small pieces (0.5 mm³). The excised gel pieces were washed with a solution containing 25 mM NH$_4$HCO$_3$ and 40% methanol, followed by 100% ACN. The proteins in gel pieces were treated with 10 mM DTT and then with 55 mM iodoacetamide. Proteins were digested with trypsin (Promega) to a final substrate: trypsin ratio of 50:1 in buffer containing 25 mM ammonium bicarbonate and 10% ACN for 12–16 h at 37 °C. The reaction was stopped by adding 5% formic acid.

For MALDI-TOF analysis, an aliquot (0.5 µL) of the supernatant from the digest was deposited onto a 384/600-µm MTP AnchorChip (Bruker Daltonik GmbH) target and allowed to air dry at room temperature. Then, an aliquot (0.5 µL) of CHCA solution (1.4 mg/ml α-Cyano-4-hydroxycinnamic acid in buffer containing 0.1% TFA, 1 mM ammonium phosphate, and 85% ACN) was added to the chip and allowed to air dry. Further MALDI-TOF MS mass spectrometry analysis was performed in positive ion mode with delayed extraction (reflection mode) on a Bruker Autoflex maX MALDI-TOF/TOF mass spectrometer (Bremen, Germany) equipped with a 200 Hz SmartBean Laser. Data acquisition and processing were done manually by using FlexControl 3.4 and Flex-Analysis 3.4 (Bruker Daltonik GmbH), respectively. The processed data were further analyzed with Biotools 3.2 (Bruker) packaged by assessing the online Mascot server to identify the corresponding polypeptides.

For tandem mass spectrometry analysis, peptides were analyzed using an UltiMate 3,000 RSLCnano UHPLC system (Thermo Fisher Scientific) equipped with a capillary C18 column (Waters, nanoEase, 130 Å, 1.7 µm, 75 µm × 250 mm). Separation was achieved with a gradient of 3% to 35% buffer B (0.1% formic acid in acetonitrile) in buffer A (0.1% formic acid in water) at a flow rate of 300 nL/min. The system was subsequently washed with 80% buffer B and reconditioned to 3% buffer B. The chromatography system was coupled to a Q-Exactive mass spectrometer (Thermo Fisher Scientific) via a Nanospray Flex Ion source. The mass spectrometer was operated in Full-MS/ddMS2 (Top N) mode, scanning a range of m/z 350–1,600 at a resolution of 70,000 FWHM with an AGC target of $3 \times 10^6$. MS2 spectra were acquired in HCD mode with a normalized collision energy of 27%, a resolution of 17,500, an injection time of 100 ms, and an AGC target

of $1 \times 10^5$. The top 10 precursors were selected for MS2 analysis using a 2.0 m/z isolation window and a dynamic exclusion time of 20 s. Raw LC–MS/MS data were analyzed using Proteome Discoverer 2.5 (Thermo Fisher Scientific). MS/MS spectra were filtered using the Spectrum Selector in Proteome Discoverer with default settings. The spectra were searched against the GyrB and ParE protein sequence databases using the SEQUEST HT and Byonic algorithms. Searches were performed in semi-specific mode to identify peptides resulting from specific cleavage at one terminus. The precursor mass tolerance was set to 10 ppm, and the fragment mass tolerance was 0.02 Da. Trypsin was used as the protease in semi-specific mode. Carbamidomethylation of cysteine was specified as a fixed modification, while N-terminal acetylation and methionine oxidation were included as variable modifications. Peptide spectrum matches (PSMs) were validated using Percolator, with a false discovery rate (FDR) threshold of 1%.

### *In vitro* protease cleavage assay

Purified protein substrates were incubated with Cpe1$_{tox}$ (wild-type or catalytic mutant) at a 10:1 ratio. Reactions were performed at 20 °C for 5 min in a final volume of 20 μL containing 20 mM Tris-HCl pH 7, 300 mM NaCl, and 1 mM DTT. The reactions were stopped by cooling on ice and the addition of an equal volume of SDS sample buffer. Samples were boiled and analyzed by SDS–PAGE, followed by Coomassie Brilliant Blue staining.

### Plasmid DNA supercoiling assay

To visualize variations of DNA topology in bacteria, we followed the procedures described in a previous study with minor modifications [112]. Overnight precultures of strains carrying pBAD33-Cpe1$_{tox}$ (wild-type or catalytic mutant) and the reporter plasmid pET were refreshed in LB broth and incubated at 37 °C with shaking (250 rpm). When cultures reached the mid-log phase, cells were transferred to a medium containing arabinose inducer and incubated for an extended period to induce Cpe1$_{tox}$ expression and intoxication. Cells were pelleted by centrifugation, and plasmid DNA was purified using the QIAprep Spin Miniprep Kit (Qiagen) and eluted by 50 mM Tris-HCl pH 8.5. DNA topoisomers were resolved on a 1% (*w/v*) agarose gel containing Tris-acetate-EDTA (TAE) and 5 μg/ml chloroquine. The gels were run in TAE buffer with 5 μg/mL chloroquine for 10 h (140 volts for 30 min, followed by 110 volts for 9.5 h), stained with 0.5 μg/mL ethidium bromide in TAE buffer, destained in distilled water overnight, and imaged with a Quantum CX5 imaging system (Vilber).

### Fluorescence and phase contrast microscopy

Imaging was performed using a DeltaVision Core Deconvolution Microscopy system with an Olympus IX71 microscope and a 60× 1.42 NA oil-immersion objective. For fluorescence and phase contrast microscopy, bacterial cultures at the mid-log phase were used to inoculate LB medium supplemented with arabinose and IPTG for the expression of Cpe1$_{tox}$ and Cpi1, respectively. After 3 h of incubation at 37 °C with shaking (200 rpm), cells were harvested by centrifugation at 18,000 × $g$ for 1 min. The culture supernatant was removed and washed with ice-cold PBS. The washed cells were resuspended in PBS containing 70% ethanol for 15-min cell fixation on ice. The fixed cells were subjected to one PBS wash. Addition of DAPI to the cell suspensions preceded a 5-min incubation on ice in the dark. The cells were then washed with PBS at 18,000 × $g$ for 1 min. The cell suspensions were then adjusted to OD$_{600}$ = 0.5 using PBS and spotted on a 1.5% (*w/v*) agarose pad settled on a microscope slide.

For time-lapse imaging of living cells, single colonies from LB agar were inoculated into LB media supplemented with appropriate antibiotics. After incubation for 16 h at 37 °C with shaking (200 rpm), the cultures were adjusted to OD$_{600}$ = 0.5 before spotting on a 1.5% (*w/v*) agarose pad containing arabinose, IPTG, and appropriate antibiotics settled on a microscope slide. Phase contrast images for each sample were acquired at 10-min intervals.

## Cpe1$_{tox}$–Cpi1/ParE$_{ATPase}$ competitive binding assay

His-tagged Cpe1$_{tox}$$^{C362A}$ protein was incubated with an excess of strep-tagged ParE$_{ATPase}$ for 1 h at 4 °C, followed by incubation with streptavidin resin (IBA Lifesciences GmbH). After binding, the resin was washed with ice-cold buffer (20 mM Tris-HCl pH 7, 300 mM NaCl, and 1 mM DTT) to remove unbound proteins. Increasing concentrations of indicated Cpi1 proteins (His-tagged) were added and incubated at 4 °C for 1 h, followed by a single wash with ice-cold buffer. Bound proteins were eluted with buffer containing 20 mM Tris-HCl pH 7, 300 mM NaCl, 1 mM DTT, and 2.5 mM desthiobiotin. Eluted samples were analyzed by SDS–PAGE, followed by Coomassie Brilliant Blue staining. Protein band intensities were quantified using ImageJ. Results from three independent experiments were plotted as a column graph.

## Biolayer interferometry (BLI)

BLI was performed using an 8-channel Gator Plus instrument (Gator Bio) with Data Acquisition V 2.15.5.1221 software (Gator Bio). Experiments were conducted at 30 °C with shaking at 1,000 rpm. We immobilized 500 nM strep-tagged Cpi1, Cpi1$^{mut4}$, or ParE$_{ATPase}$ onto strep-tactin XT-coated biosensor tips (IBA Lifesciences). Equilibrium binding dose-response curves were generated by varying the concentrations of His-tagged Cpe1$_{tox}$$^{C362A}$ in BLI buffer (0.2% (*w/v*) BSA, 0.02% (*v/v*) Tween-20 in PBS). For Cpi1 and Cpi1$^{mut4}$, the association and dissociation intervals were conducted 150 and 300 s, whereas both intervals were set as 120 s for ParE$_{ATPase}$. BLI data analysis included reference channel subtraction and Savitzky–Golay filtering. Kinetic parameters were determined using a 1:1 global fitting model with $R_{max}$ values linked across sensors. The interaction between ParE$_{ATPase}$ and His-tagged Cpe1$_{tox}$$^{C362A}$ exhibited fast-on-fast-off kinetics. Therefore, steady-state analysis of the response curves was employed to calculate the $K_d$ and nominal $R_{max}$ values. All data represent results from at least three independent experiments.

## Generation of sequence logos

Protein sequences of 35,812 GyrB homologs and 14,212 ParE homologs were retrieved from the UniProt database [113]. Regions containing the consensus sequence (LHAGGKF) were extracted and aligned using ClustalW [96]. The results were visualized as a sequence logo using the WebLogo3 online tool [114]. Sequences flanking the substrate cleavage sites were obtained from previous studies to generate the sequence logo for the consensus sequences on the substrates of ComA and LahT [50,60]. Sequence conservation was analyzed according to the same ClustalW alignment and visualization procedures.

## Quantification and statistical analysis

Statistical significance in bacterial co-culture and competition assays was assessed by unpaired two-tailed *t*-tests between relevant samples. Protein levels on SDS–PAGE were compared between groups by paired two-tailed *t*-tests. All statistical analyses were performed using Prism version 10.3.0 for windows, GraphPad Software, Boston, Massachusetts USA (www.graphpad.com). Exact *p*-values indicating statistical significance are provided on the graphs.

## Supporting information

**S1 Fig. Interbacterial PLCP effectors and their cognate immunity proteins are distributed across various gram-negative bacteria.** Related to Fig 1b. **(a)** Multiple sequence alignment of 155 unique sequences of interbacterial PLCP effectors. Sequences have been ordered by relative identity to Cpe1$_{tox}$ of *E. coli* ATCC-11775 and are displayed in two clusters divided according to the associated secretion systems (T6SS or CDI). Conserved residues are indicated as follows: the catalytic triad residues are highlighted by red (Cysteine) and orange (Histidine and Aspartate/Asparagine) shadows; and hydrophobic residues composing the surface concave region are highlighted by blue shadows. The Phylum of each strain containing the interbacterial PLCP effectors is indicated by colored boxes (as indicated in the legend).

**(b)** Sequences of the putative cognate immunity proteins related to the interbacterial PLCP effectors in **(a)**. Conserved binding residues are highlighted in green shadows. Blank rows indicate an absence of putative immunity proteins due to incomplete genomic data for certain bacterial species in the NCBI database. See also S1 Table for detailed information. (TIF)

**S2 Fig. Interbacterial PLCP effectors exhibit no correlations in phylogeny, associated secretion pathways, or taxonomy.** Related to Fig 1d. Maximum likelihood phylogeny of 155 unique PLCP effector sequences visualized as a rectangle tree. Branch support values (SH-aLRT and ultrafast bootstrap) are displayed as symbols (filled diamond: SH-aLRT ≥ 80%, hollow diamonds: ultrafast bootstrap ≥ 95%). Tips representing sequences of the closely related house-keeping PLCPs, ComA$_{pep}$ and LahT$_{pep}$, as well as the previously identified interbacterial PLCP effectors EvpQ and TsaP, are highlighted in red. The taxonomy of strains containing the PLCP effectors and their associated secretion pathways is indicated on the right as colored boxes. The scale bar represents the average number of substitutions per site. See also S1 Table for detailed information. The original tree file used to generate this figure is available as S2 Data. (TIF)

**S3 Fig. Purification of the Cpe1$_{tox}$–Cpi1 complex and protein expression levels of Cpi1 variants.** Related to Fig 2. **(a)** Coomassie-stained SDS–PAGE analysis of the His-tagged Cpe1$_{tox}$ co-purified with Cpi1. **(b)** Protein expression levels of wild-type Cpi1 and variants, as assessed by immunoblotting analysis. The cytosolic protein RpoB was used as a loading control. The original images are available in S1 Raw Images. (TIF)

**S4 Fig. ComA$_{pep}$ and LahT$_{pep}$ feature hydrophobic concave regions on their surfaces near the catalytic pocket.** Related to Fig 3. **(a, b)** Depiction of ComA$_{pep}$ **(a)** and LahT$_{pep}$ **(b)** illustrating the location of their respective catalytic pockets and hydrophobic concave surfaces. ComA$_{pep}$ (PDB: 3K8U, light gray). LahT$_{pep}$ (PDB: 6MPZ, dark gray). (TIF)

**S5 Fig. Cpe1$_{tox}$ specifically targets and cleaves GyrB and ParE among eight candidate substrates identified from the interactome analysis.** Related to Fig 3. **(a-f)** Coomassie-stained SDS–PAGE analysis of the *in vitro* cleavage assay of six candidate substrates of Cpe1$_{tox}$: AccA **(a)**, DapE **(b)**, FtsZ **(c)**, NusA **(d)**, NusG **(e)**, and RpoB **(f)**. **(g, h)** Presence of Cpi1 suppressed the cleavage of GyrB **(g)** and ParE **(h)** by Cpe1$_{tox}$. The substrates were incubated with Cpe1$_{tox}$ (lanes 1 and 2), Cpe1$_{tox}$$^{C362A}$ (lanes 3 and 4), or Cpe1$_{tox}$ and Cpi1 (lanes 5 and 6). Cleaved fragments are indicated with red arrow-heads. The original images are available in S1 Raw Images. (TIF)

**S6 Fig. Cleavage sequences of ComA and LahT and mass spectrometric analysis of Cpe1-cleaved fragments.** Related to Fig 4. **(a, b)** Sequence logos showing the consensus sequence recognized by ComA **(a)** and LahT **(b)**. **(c, d)** Mapping of peptide sequences from Cpe1$_{tox}$-digested GyrB **(c)** and ParE **(d)** fragments. Cpe1$_{tox}$-cleaved fragments, resolved by Coomassie-stained SDS–PAGE, were purified, trypsin-digested, and subjected to MALDI-TOF analysis. The intensity of the signals (*Y*-axis) from identified peptides and their coverage across the respective full-length protein (*X*-axis) are plotted below. Upper chart: peptides from the small fragment. Lower chart: peptides from the large fragment. **(e, f)** Tandem mass spectrum of indicated peptides from Cpe1$_{tox}$-cleaved GyrB **(e)** and ParE **(g)** fragments. Fragmentation ions (b, blue; y, red) with resolved spectra and the residues correlating to the LHAGGKF motif (bold) are indicated. The quantitative data for generating plots can be found in S1 Data. The raw proteomics data can be found in https://doi.org/10.5281/zenodo.15361709. (TIF)

**S7 Fig. Cpe1 intoxication inhibits chromosome segregation in *E. coli*.** Related to Fig 4f. **(a)** Phase-contrast (top), blue fluorescence (middle), and merged (bottom) images of *E. coli* carrying an empty vector, *E. coli* expressing Cpe1$_{tox}$, *E.*

*coli* expressing Cpe1$_{tox}$$^{C362A}$, or *E. coli* co-expressing Cpe1$_{tox}$ and Cpi1, after three hours of induction. Scale bar = 20 μm. The white borders demarcate the cropped images displayed in Fig 4f.
(TIF)

**S8 Fig. Z-ring formation is disrupted in Cpe1-intoxicated *E. coli*.** Related to Fig 4f. **(a)** Phase-contrast (top), blue fluorescence (second from top), green fluorescence (third from top), and merged (bottom) images of *E. coli* carrying an empty vector, shown before and after 2-h incubation. **(b)** Fluorescence micrographs of *E. coli* intoxicated by Cpe1$_{tox}$. Phase-contrast (top), blue fluorescence (second from top), green fluorescence (third from top), and merged (bottom) images are presented. White borders indicate the zoomed-in regions shown in the bottom-right corner of each image. Scale bar = 20 μm.
(TIF)

**S9 Fig. Interactions between Cpe1$_{tox}$ with the substrates and kinetics of Cpe1$_{tox}$ with Cpi1$^{mut4}$.** Related to Fig 5a, 5c, 5d, 5e, 5f. **(a, b)** Predicted assigned error (PAE) plots for the models of the Cpe1$_{tox}$–GyrB$_{ATPase}$ interaction **(a)** and the Cpe1$_{tox}$–ParE$_{ATPase}$ interaction **(b)** in Fig 5a. Accuracy of the predicted relative positions of subunits within the complex, as indicated by AlphaFold 3 prediction scores (iPTM). Confidence in the overall folding of the complex is indicated by pTM scores. **(c)** Predicted structure of Cpe1$_{tox}$–GyrB$_{ATPase}$ interaction, showing the entrance of the consensus loop of GyrB into the active site of Cpe1 (left panel). The magnified view of the active pocket (right panel) implies possible catalysis. **(d)** Cpe1$_{tox}$ targets the ATPase domain of ParE. Cleavage of full-length ParE and the ATPase domain by Cpe1$_{tox}$ was analyzed by Coomassie-stained SDS–PAGE. Cleaved fragments are indicated with red arrowheads. **(e, f)** Quantifications of the results of the competitive binding assay shown in Fig 5c **(e)** and 5d **(f)**. Results from three independent assays were plotted on a column graph as mean ± SD. The *Y*-axis represents levels of Cpe1$_{tox}$ normalized to the level of ParE$_{ATPase}$ on the same lane (%). The *X*-axis represents the concentration of wild-type or mutant Cpi1 in the reaction. *P*-values were calculated using Student *t* test to compare each result from that of no competition binding (0 μM Cpi1). **(g)** Results of kinetic assays on interactions between Cpi1$^{mut4}$ and Cpe1$_{tox}$$^{C362A}$ using biolayer interferometry (BLI). Data from three replicates have been plotted on an *X–Y* scatter graph. The *Y*-axis represents wavelength shifts (nm) generated by the binding of the two proteins, which are nearly non-detectable (N.D.). The *X*-axis represents reaction time in seconds. A vertical dashed line marks the transition from the association step to the dissociation step. Kinetic parameters such as $K_d$, $K_{on}$, $K_{off}$, and $R_{max}$ could not be calculated and are labeled as ND. The data underlying this figure are available in S1 Data and S1 Raw Images.
(TIF)

**S10 Fig. Overexpression of cleavage-resistant mutant type II DNA topoisomerases (GyrB$^{ΔG102}$ and ParE$^{ΔG98}$) does not alleviate Cpe1 toxicity. (a, b)** Expression level of Strep-tagged GyrB$^{ΔG102}$ **(a)** and His-tagged ParE$^{ΔG98}$ **(b)** in *E. coli* susceptible to Cpe1. **(c)** Viable cells recovered from plating cultures carrying plasmids expressing the indicated proteins. *E. coli* carrying plasmid-borne GyrB$^{ΔG102}$ and ParE$^{ΔG98}$ or an empty vector was included to validate suppression of Cpe1 toxicity. **(d)** Bacterial competition assays were performed between the indicated donor cell carrying Cpe1 and the susceptible strain. Experiments were conducted on a solid agar supplemented with inducing agents to maintain expression of plasmid-borne GyrB$^{ΔG102}$ and ParE$^{ΔG98}$ in the recipient strain. The data in **(c)** and **(d)** are shown as the mean ± SD; *n* = 6. Both data are representative experiments out of at least 3 independent experiments. *P* values were calculated with Student *t* test to assess differences in viability among populations **(c)**, and to evaluate statistically significant differences in the competitive indices of each donor strain against the specified recipients **(d)**. The data underlying this figure are available in S1 Data and S1 Raw Images.
(TIF)

**S1 Video. Time-lapse phase contrast microscopy series of *E. coli* containing an empty vector. Related to Fig 2.**
(AVI)

**S2 Video. Time-lapse phase contrast microscopy series of *E. coli* expressing Cpe1$_{tox}$. Related to** [Fig 2](Fig 2)**.**
(AVI)

**S3 Video. Time-lapse phase contrast microscopy series of *E. coli* expressing Cpe1$_{tox}^{C362A}$. Related to** [Fig 2](Fig 2)**.**
(AVI)

**S4 Video. Time-lapse phase contrast microscopy series of *E. coli* co-expressing Cpe1$_{tox}$ and Cpi1. Related to** [Fig 2](Fig 2)**.**
(AVI)

**S1 Table. Identified interbacterial PLCP effectors and their putative cognate immunity proteins.** Relevant to [Figs 1](Figs 1), [S1](S1), and [S2](S2).
(XLSX)

**S2 Table. X-ray data collection and refinement statistics.** Relevant to [Methods](Methods) and [Fig 2](Fig 2).
(AVI)

**S3 Table. The interactome of Cpe1 identified by Bpa-crosslinking and LC–MS/MS analysis.** Relevant to [Fig 3](Fig 3).
(XLSX)

**S4 Table. Strains, recombinant DNA, and oligonucleotides used in this study.**
(XLSX)

**S1 Data. Raw data for figures containing numerical data.**
(XLSX)

**S2 Data. The original tree file for generating the phylogenetic trees in** [Figs 1D](Figs 1D) **and** [S2](S2)**.**
(TREEFILE)

**S1 Raw Images. The uncropped western blots and gel electrophoresis images shown in figures can be found in this dataset.**
(PDF)

## Acknowledgments

We thank Drs. Harmit Malik, Dor Salomon, Jun-Yi Leu, Erh-Min Lai, Chih-Horng Kuo, Chih-Yen King, and Ting laboratory members for helpful discussions; Drs. Joseph Mougous, Ming-Yang Ho, Hsin-Hung Chou, and Sue Lin-Chao for sharing reagents; Dr. Hung-Ta Chen for sharing equipment; Drs. Hsin-Nan Lin, Chen-Hsin Yu, Yae-Huei Liou, Sue-Ping Lee, Yung-Hsuan Wu, Shu-Yu Lin, Chuan-Chih Hsu, Chin-Wen Chen, Shu-Chuan Jao, Jin-Hsuan Yu, and Xin-Jie Huang for technical assistance; and Dr. John O'Brien for manuscript editing.

## Author contributions

**Conceptualization:** Pin-Yi Song, Chia-En Tsai, Chuan Ku, Kuo-Chiang Hsia, See-Yeun Ting.

**Data curation:** Pin-Yi Song, Chia-En Tsai, Yung-Chih Chen, Yu-Wen Huang, Po-Pang Chen, Tzu-Haw Wang, Chao-Yuan Hu, Po-Yin Chen, See-Yeun Ting.

**Formal analysis:** Pin-Yi Song, Chia-En Tsai, See-Yeun Ting.

**Writing – original draft:** Pin-Yi Song, Chia-En Tsai, Po-Yin Chen, See-Yeun Ting.

**Writing – review & editing:** Po-Yin Chen.

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
