## [Editor Report · Decision Letter 0]

1 Apr 2025

Dear Dr Ting, 

Thank you for submitting your manuscript entitled "An Interbacterial Cysteine Protease Toxin Inhibits Cell Growth by Targeting Type II DNA Topoisomerases GyrB and ParE" for consideration as a Research Article by PLOS Biology.

Your manuscript has now been evaluated by the PLOS Biology editorial staff, as well as by an academic editor with relevant expertise, and I am writing to let you know that we would like to call in your study but send your revised manuscript back to the original reviewers from Review Commons for their feedback.

Once your full submission is complete, your paper will undergo a series of checks in preparation for peer review. After your manuscript has passed the checks it will be sent out for re-review. To provide the metadata for your submission, please Login to Editorial Manager (https://www.editorialmanager.com/pbiology) within two working days, i.e. by Apr 03 2025 11:59PM.

Kind regards,

Melissa

Melissa Vazquez Hernandez, Ph.D.

Associate Editor

PLOS Biology

---

## [Decision Letter · Decision Letter 1]

21 Apr 2025

Dear Dr Ting,

Thank you for your patience while we considered your revised manuscript "An Interbacterial Cysteine Protease Toxin Inhibits Cell Growth by Targeting Type II DNA Topoisomerases GyrB and ParE" for consideration as a Research Article at PLOS Biology. Your revised study has now been evaluated by the PLOS Biology editors, the Academic Editor and the original reviewers. 

In light of the reviews, which you will find at the end of this email, we are pleased to offer you the opportunity to address the remaining points from the reviewers in a revision that we anticipate should not take you very long. We will then assess your revised manuscript and your response to the reviewers' comments with our Academic Editor aiming to avoid further rounds of peer-review, although we might need to consult with the reviewers, depending on the nature of the revisions.

In light of the reviews, which you will find at the end of this email, while Reviewers 2 and 3 have agreed that the manuscript can be accepted as it is, Reviewer 1 is still raised an important concern. Reviewer 1 is not completely convinced about the explanation that overexpression of cleavage-resistant mutants of GyrB and ParE cannot confer resistance to Cpe1, and would like you to show immunoblotting of the expression of these mutants. We think that this request is reasonable and will ask you to include it in your revision. 

We expect to receive your revised manuscript within 2 months. Please email us (plosbiology@plos.org) if you have any questions or concerns, or would like to request an extension. 

**IMPORTANT - SUBMITTING YOUR REVISION**

*Resubmission Checklist*

*Published Peer Review*

*PLOS Data Policy*

*Blot and Gel Data Policy*

Sincerely,

Melissa

Melissa Vazquez Hernandez, Ph.D.

Associate Editor

PLOS Biology

Reviewer #1: 

The results that overexpression of cleavage-resistant mutants of GyrB and ParE cannot confer resistance to Cpe1 in both co-expression and killing assays is unexpected, particularly when these mutants were expressed from multi-copy plasmids. The authors' explanation was not satisfactory. Did the authors examine the expression of these mutants by methods like immunoblotting? If these mutants were properly expressed in the testing strains, and these strains are still resistant to Cpe1-meidated killing, the results should be included in the paper and appropriately discussed to acknowledge that Cpe1 likely targets other essential proteins.

Reviewer #2 (Christina R Bourne): 

I commend the authors on their careful revision in response to previous reviewer comments, including my own. Overall I believe this is a very well executed study of high interest to multiple sub-areas in microbiology. In my opinion, the revised version is acceptable without further changes.

Reviewer #3: 

This is a paper that I earlier reviewed and the authors have satisfactorily addressed all my comments

---

## [Editor Report · Decision Letter 2]

30 Apr 2025

Dear Dr Ting,

Thank you for your patience while we considered your revised manuscript "An Interbacterial Cysteine Protease Toxin Inhibits Cell Growth by Targeting Type II DNA Topoisomerases GyrB and ParE" for publication as a Research Article at PLOS Biology. This revised version of your manuscript has been evaluated by the PLOS Biology editors, and the Academic Editor.

Based on our Academic Editor's assessment of your revision, we are likely to accept this manuscript for publication, provided you satisfactorily address the remaining editorial points. Please also make sure to address the following data and other policy-related requests.

a) Thank you so much for already providing all raw data from the figures. I just noticed that the labeling in the excel sheet for Figure 5 says "3E" and "3F". 

b) Please cite the location of the data clearly in all relevant main and supplementary Figure legends, e.g. “The data underlying this Figure can be found in S1 Data” or “The data underlying this Figure can be found in https://doi.org/10.5281/zenodo.XXXXX”

c) Please provide the tree files for the phylogenetic trees in Figures 

d) Please provide the raw proteomics data for Figure S6EF by uploading them in a repository like Zenodo, or proteomics-specific repositories like PRIDE. You can read about our policies and recommendations for repositories here: https://journals.plos.org/plosbiology/s/recommended-repositories

e) Please ensure that your Data Statement in the submission system accurately describes where your data can be found and is in final format, as it will be published as written there.

f) Per journal policy, if you have generated any custom code during the course of this investigation, please make it available without restrictions upon publication. Please ensure that the code is sufficiently well documented and reusable, and that your Data Statement in the Editorial Manager submission system accurately describes where your code can be found.

We expect to receive your revised manuscript within two weeks. 

*Published Peer Review History*

*Press*

Sincerely,

Melissa

Melissa Vazquez Hernandez, Ph.D.

Associate Editor

PLOS Biology

---

## [Editor Report · Decision Letter 3]

12 May 2025

Dear Dr Ting,

Thank you for the submission of your revised Research Article "An Interbacterial Cysteine Protease Toxin Inhibits Cell Growth by Targeting Type II DNA Topoisomerases GyrB and ParE" for publication in PLOS Biology. On behalf of my colleagues and the Academic Editor, Michael Laub, I am pleased to say that we can in principle accept your manuscript for publication, provided you address any remaining formatting and reporting issues. These will be detailed in an email you should receive within 2-3 business days from our colleagues in the journal operations team; no action is required from you until then. Please note that we will not be able to formally accept your manuscript and schedule it for publication until you have completed any requested changes.

PRESS

Sincerely, 

Melissa

Melissa Vazquez Hernandez, Ph.D., Ph.D.

Associate Editor

PLOS Biology
